# A full gap above the Fermi level: the charge density wave of monolayer VS$_2$

Camiel van Efferen [1✉], Jan Berges [2], Joshua Hall[1], Erik van Loon [2], Stefan Kraus[1], Arne Schobert[2], Tobias Wekking[1], Felix Huttmann[1], Eline Plaar[1], Nico Rothenbach[3], Katharina Ollefs [3], Lucas Machado Arruda [4], Nick Brookes[5], Gunnar Schönhoff[2], Kurt Kummer [5], Heiko Wende [3], Tim Wehling [2,6] & Thomas Michely[1]

In the standard model of charge density wave (CDW) transitions, the displacement along a single phonon mode lowers the total electronic energy by creating a gap at the Fermi level, making the CDW a metal–insulator transition. Here, using scanning tunneling microscopy and spectroscopy and ab initio calculations, we show that VS$_2$ realizes a CDW which stands out of this standard model. There is a full CDW gap residing in the unoccupied states of monolayer VS$_2$. At the Fermi level, the CDW induces a topological metal-metal (Lifshitz) transition. Non-linear coupling of transverse and longitudinal phonons is essential for the formation of the CDW and the full gap above the Fermi level. Additionally, x-ray magnetic circular dichroism reveals the absence of net magnetization in this phase, pointing to coexisting charge and spin density waves in the ground state.

[1] II. Physikalisches Institut, Universität zu Köln, Zülpicher Straße 77, 50937 Köln, Germany. [2] Institut für Theoretische Physik, Bremen Center for Computational Materials Science, and MAPEX Center for Materials and Processes, Universität Bremen, Otto-Hahn-Allee 1, 28359 Bremen, Germany. [3] Fakultät für Physik und Center für Nanointegration Duisburg-Essen (CENIDE), Universität Duisburg-Essen, Carl-Benz-Straße, 47057 Duisburg, Germany. [4] Institut für Experimentalphysik, Freie Universität Berlin, Arnimallee 14, 14195 Berlin, Germany. [5] European Synchrotron Research Facility (ESRF), Avenue des Martyrs 71, CS 40220, 38043 Grenoble Cedex 9, France. [6] Institute of Theoretical Physics, Universität Hamburg, Notkestraße 9-11, 22607 Hamburg, Germany. ✉email: efferen@ph2.uni-koeln.de

The many-body ground states of two-dimensional (2D) materials, wherein the reduced dimensionality leads to the enhancement of correlation effects, have been extensively researched in recent years. Of particular interest are the coexistence or competition between charge density waves (CDWs), as found in many 2D transition metal dichalcogenides (TMDCs), with superconducting and magnetic phases[1,2]. Since these phases can be strongly dependent on the substrate[3,4] or the defect density[5,6], the intrinsic properties of 2D materials are difficult to determine experimentally. In addition, CDWs themselves are the subject of an ongoing controversy regarding the driving force behind the CDW transition and the exact structure of the electronic system in the CDW phase of 2D materials[7,8].

Peierls' explanation for the CDW in a one-dimensional chain of atoms states that periodic lattice distortions open an electronic gap at the nesting wavevector. This gap at the Fermi level lowers the energy of the occupied states and thus the total energy, while increasing the energy of the unoccupied states that do not contribute to the total energy. Thus, this gapping mechanism requires the gap to be at the Fermi level. However, in many (quasi-)2D cases, CDWs form in the complete or partial absence of Fermi-surface nesting, suggesting that the driving mechanism behind their formation lies beyond a simple electronic disturbance[9], and it has been questioned whether the concept of nesting is essential for understanding CDW formation[10–12]. Instead, a strong and wavevector-dependent electron–phonon coupling is often predicted to be the driving force behind the transition[7]. For these CDWs, spectral reconstructions are not limited to a small energy window around the Fermi energy, but can occur throughout the entire electronic structure, opening the door to novel spectral fingerprints of the CDW. A full gap could occur away from the Fermi energy. However, even for the well-studied strong-coupling TMDCs 2H-NbSe$_2$[9,13,14] and 1T-TaS$_2$[7,15–17], no experimental verification of a clear CDW gap located away from the Fermi energy has been provided to date. Furthermore, at the Fermi energy, the undistorted phase and the CDW can have different Fermi-surface topologies, with the implication that the transition is a metal–metal Lifshitz transition[18].

Metallic 1T-VS$_2$ is not only a promising electrode material in lithium-ion batteries[19,20], but also a prototypical $d^1$ system, expected to host strongly correlated physics[21]. It is stated to be a CDW material[22,23] and a candidate for 2D magnetism[24,25] with layer-dependent properties[26], making it a model system for investigating complex ground states. Although difficult to synthesize, bulk 1T-VS$_2$ has been well studied, with many authors finding a CDW transition at around 305 K when it was prepared via the de-intercalation of Li[22,23,27–29]. However, recent powder samples prepared under high pressure show no CDW transition[30]. Based on their finding of a phonon instability at 2/3 $\overline{\Gamma K}$ corresponding to the experimental CDW wavevector of Li de-intercalated bulk samples[23], Gauzzi et al. point out that bulk "VS$_2$ is at the verge of CDW transition"[30] but not a CDW material. Due to a similar difficulty in synthesis, the properties of monolayer 1T-VS$_2$ have proven equally elusive[31]. Theoretical calculations had predicted ferromagnetism and a CDW with a wavevector of 2/3 $\overline{\Gamma K}$[21,25]. When it was first synthesized however, scanning tunneling microscope (STM) measurements did not reveal a CDW[31], presumably due to strong hybridization with the Au(111) substrate, similar to the case of 2H-TaS$_2$ on Au(111)[4,32,33].

Here we report the growth of VS$_2$ monolayers on the inert substrate graphene (Gr) on Ir(111) via a two-step molecular beam epitaxy (MBE) synthesis developed for sulfur-based TMDCs[34]. Using a combination of STM, scanning tunneling spectroscopy (STS), and ab initio density functional theory (DFT) calculations, we determine the spatial and electronic structure of monolayer

VS$_2$. We observe a $\mathbf{q} \approx 2/3$ $\overline{\Gamma K}$ CDW as the electronic ground state at 7 K, which remains stable up to room temperature. A full gap in the density of states (DOS), residing completely in the unoccupied states, is measured via STS. From DFT and density functional perturbation theory (DFPT), we find that, although a transverse phonon mode initially becomes unstable in the harmonic approximation, the final CDW has a substantial admixture of longitudinal modes. The calculations are in excellent agreement with experiment, regarding both the electronic structure of the CDW phase and the spatial charge distribution observed on the VS$_2$ islands.

X-ray magnetic circular dichroism (XMCD) measurements at 7 K and 9 T robustly show vanishing total net magnetization. The coupling of the CDW to a spin density wave (SDW), energetically favored in DFT calculations, could explain this observation, providing interesting prospects for future research on the interplay of CDWs and magnetism.

## Results

**CDW in monolayer VS$_2$.** The typical morphology of the MBE-grown monolayer VS$_2$ islands on Gr/Ir(111) is shown in the large-scale STM image in Fig. 1a. The islands were grown by room-temperature deposition of vanadium in a sulfur background pressure of $P_S^g = 1 \times 10^{-8}$ mbar and subsequently annealed at 600 K in the same sulfur pressure. Annealing to temperatures of 800 K and above leads to the formation of a variety of sulfur-depleted phases, which are not under concern here. Similar observations were made by Arnold et al.[31], who established monolayer stoichiometric 1T-VS$_2$ on Au(111) by annealing in a sulfiding gas at 670–700 K, while sulfur-depleted monolayer phases form when annealed to the same or higher temperature in the absence of sulfiding species. We also note that depending on growth temperature and sulfur pressure bilayer samples without any monolayer islands evolve.

The monolayer islands are fully covered by a striped superstructure which is present regardless of island size or defect density and occurs in domains, typically separated by grain boundaries. In the topograph of Fig. 1b, taken at 7 K, the VS$_2$ lattice is resolved, exhibiting the hexagonal arrangement of top layer sulfur atoms as protrusions. We find that monolayer VS$_2$ has a lattice constant of $a_{VS_2} = (3.21 \pm 0.02)$ Å, in good agreement with the bulk lattice constant of 3.22 Å of 1T-VS$_2$[27,35]. The similarity of the lattice constants indicates also the absence of epitaxial strain, consistent with the random orientation of the VS$_2$ with respect to the Gr.

The stripes of the superstructure have an average periodicity of $(2.28 \pm 0.02)a_{VS_2}$. Close analogues to this structure have previously been observed in stoichiometric monolayer VSe$_2$. There, a superstructure of identical symmetry is attributed to a CDW[3,36–38] [compare Supplementary Fig. 1]. The superstructure is found to persist up to room temperature, as can be concluded from the STM topograph in Fig. 1c, taken at 300 K. At this temperature, the superstructure appears spontaneously only on larger islands, suggesting that the transition temperature between the superstructure and the undistorted phase is not far above room temperature. Indeed, on smaller islands the STM tip can be used to reversibly switch between the undistorted $(1 \times 1)$ structure and the superstructure, shown in Supplementary Fig. 2. This directly excludes the possibility that the superstructure is due to a sulfur-depleted phase. We conclude that the superstructure is most likely a CDW in a stoichiometric monolayer of 1T-VS$_2$.

For the DFT calculations below, the experimental wave pattern must be approximated by a commensurate structure. A close approximation with periodicity $2.25a_{VS_2}$ is overlaid on the atomic resolution image in Fig. 1b. It locally matches the

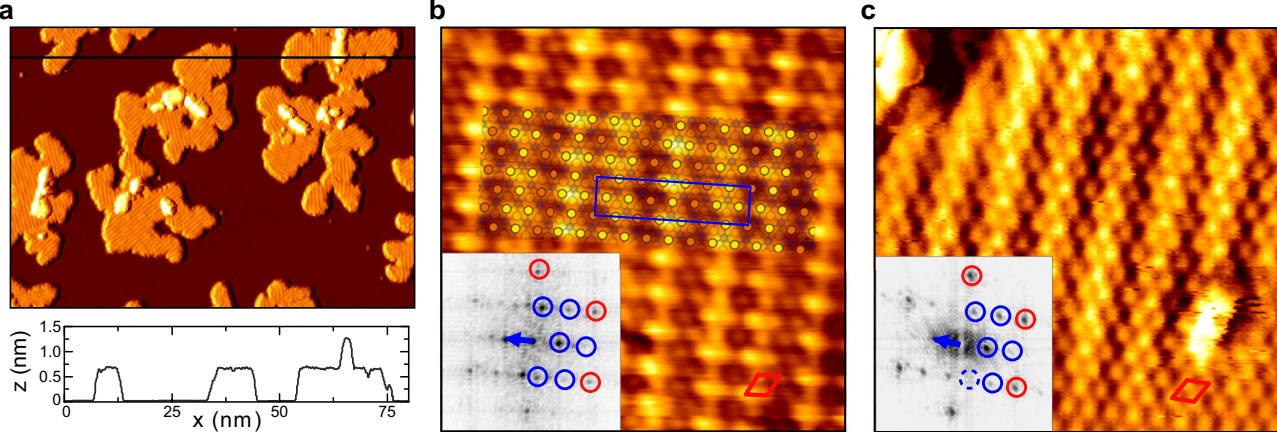

**Fig. 1 Structure of VS$_2$ on Gr/Ir(111) at 7 and 300 K. a** Large-scale 7 K STM topograph of monolayer VS$_2$ islands with small bilayers present. A height profile along the horizontal black line is shown below the image. **b, c** Atomically resolved STM images of monolayer VS$_2$ at 7 K (**b**) and 300 K (**c**). The Fourier transform of each image is shown as an inset, with the 1 × 1 VS$_2$ structure in red and the superstructure spots indicated in blue. In **b**, an atomic model for the 9× √3R30° superstructure is included as an overlay. The model depicts the top sulfur atoms, with their apparent height in STM coded in orange (low) and yellow (high). Measurement parameters: **a** 80 × 50 nm$^2$, $I_t$ = 0.8 nA, $V_t$ = −800 meV, **b** 6 × 6 nm$^2$, $I_t$ = 0.6 nA, $V_t$ = 400 meV, **c** 6 × 6 nm$^2$, $I_t$ = 1.0 nA, $V_t$ = −1000 meV.

incommensurate CDW quite well. The blue box indicates the corresponding 9× √3R30° unit cell. The Fourier transform of the topograph is shown as inset in Fig. 1b, with the wavevector of the CDW indicated (blue arrow). The same is done for the 300 K topograph in Fig. 1c. Within the margin of error, the wavevector is found to be temperature independent, with $\mathbf{q}_{CDW(7K)}$ = $(0.656 \pm 0.006)$ $\overline{\Gamma K}$ and $\mathbf{q}_{CDW(300K)}$ = $(0.65 \pm 0.03)$ $\overline{\Gamma K}$. Since the wavevector of the 9× √3R30° unit cell, $\mathbf{q}_{9 \times \sqrt{3}R30°}$ = $2/3$ $\overline{\Gamma K} \approx 0.667$ $\overline{\Gamma K}$, is slightly larger than the experimental value, we will in the following also consider another unit cell of size 7× √3R30°, with a slightly smaller wavevector $\mathbf{q}_{7 \times \sqrt{3}R30°}$ = $9/14$ $\overline{\Gamma K} \approx 0.643$ $\overline{\Gamma K}$. With the experimental wavevector lying in between $\mathbf{q}_{7 \times \sqrt{3}R30°}$ and $\mathbf{q}_{9 \times \sqrt{3}R30°}$, calculations with these two unit cells should capture the essential features of the incommensurate structure and provide a check on any artefacts or errors arising from using them for computational purposes (cf. Supplementary Fig. 3).

**Energetics of lattice instabilities.** Ab initio DFPT calculations of the acoustic phonon dispersion of undistorted monolayer 1T-VS$_2$ confirm that a structural instability and corresponding tendencies toward CDW formation exist for the experimental wavevector. Figure 2a shows that the longitudinal–acoustic and transverse–acoustic modes feature imaginary frequencies in several parts of the Brillouin zone. In other words, the Born–Oppenheimer energy surface is a downwards-opening parabola for small atomic displacements in the direction of these modes, as seen in Fig. 2b (triangle marks). At the experimental wavevector between $\mathbf{q} = 2/3$ $\overline{\Gamma K}$ and $\mathbf{q} = 9/14$ $\overline{\Gamma K}$, we find an instability of the transverse–acoustic branch. However, the dominant instability within the harmonic approximation (i.e., DFPT), is located at $\mathbf{q} = 1/2$ $\overline{\Gamma M}$ in the longitudinal–acoustic branch.

To go beyond the harmonic approximation, we have performed structural relaxations on appropriate unit cells. The resulting atomic positions are shown in Fig. 2c–e. On the aforementioned 9× √3R30° and 7× √3R30° unit cells, which can approximately host an integer multiple of the observed wavelength, the vanadium atoms are displaced from their symmetric positions by up to 8% of the lattice constant, while

the positions of the sulfur atoms remain almost unchanged, see Fig. 2c, d. The associated energy gains amount to about 23 meV per VS$_2$ formula unit (cf. ref. [21]). The magnitude of these distortions and energy gains is similar to other octahedral TMDCs but exceeds by far what is found in trigonal–prismatic TMDCs[7,39]. For instance, on the DFT level, the maximum displacement in the √13× √13 CDW of 1T-NbSe$_2$ is 8.8% of the lattice constant with an energy gain of 57 meV per formula unit[40], while in the 3 × 3 CDW of 2H-NbSe$_2$ distortions and energy gain amount to only 2.3% of the lattice constant and 3.7 meV per formula unit[41].

The vanadium displacement has components in both the transverse and longitudinal direction (vertical and horizontal in Fig. 2c, d), even though the instability of the phonons at $\mathbf{q} = 2/3$ $\overline{\Gamma K}$ and $\mathbf{q} = 9/14$ $\overline{\Gamma K}$ is of transverse character (white arrows in Fig. 2c, d). As a consequence, all longitudinal displacement components must stem from non-linear mode–mode coupling beyond the harmonic approximation. The admixture of longitudinal displacement components stems mainly from wavevectors $\mathbf{q} = 4/3$ $\overline{\Gamma K}$ and $\mathbf{q} = 9/7$ $\overline{\Gamma K}$, which are also commensurate with the 9× √3R30° and the 7× √3R30° unit cells, respectively. The admixed longitudinal modes at $\mathbf{q} = 4/3$ $\overline{\Gamma K}$ and $\mathbf{q} = 9/7$ $\overline{\Gamma K}$ are stable in the harmonic approximation and the non-linear admixture is not related to any nesting or Peierls physics (cf. Supplementary Fig. 4e, f).

We also find a distorted ground state on a 4 × 4 unit cell, see Fig. 2e. This structure is commensurate to the six wavevectors $\mathbf{q} = 1/2$ $\overline{\Gamma M}$, where we have instabilities in the longitudinal–acoustic branch arising from near perfect Fermi-surface nesting, see Supplementary Fig. 4a. However, here the displacements amount to only 4% of the lattice constant with a corresponding energy gain below 4 meV per 1T-VS$_2$ formula unit —much less than what is found for the 7× √3R30° or 9× √3R30° CDW structures. Thus, the DFT total energies of the fully relaxed structures are in line with the experimentally observed CDW patterns.

To illustrate the significance of the non-linear mode–mode coupling, in Fig. 2b, we also show the Born–Oppenheimer energy surfaces for displacements toward the relaxed structures (circle marks). The energy curve of the 4 × 4 structure is steeper in the vicinity of the origin. In other words, the 4 × 4 structure wins for

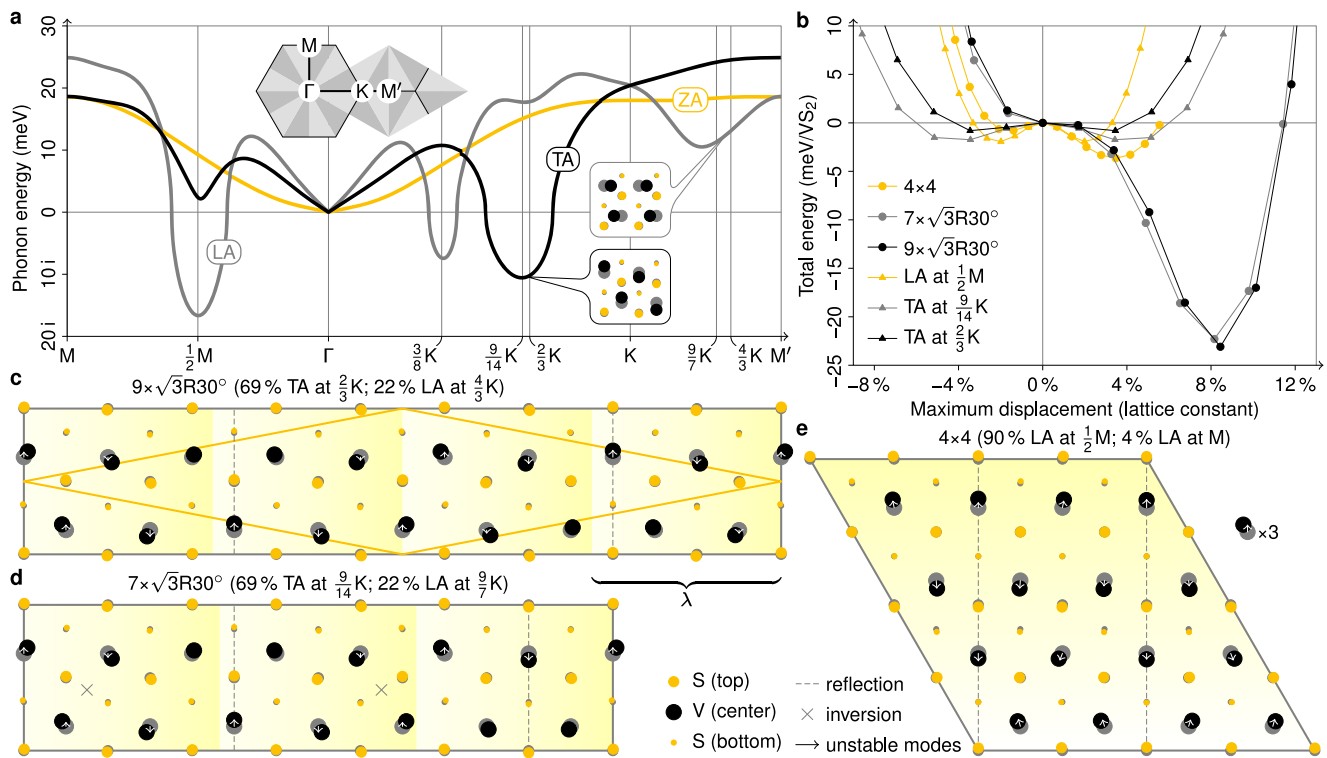

**Fig. 2 Lattice instabilities in monolayer 1T-VS₂ from first principles. a** Acoustic phonon dispersion from DFPT. LA, TA, and ZA stand for dominant longitudinal, transverse, and out-of-plane atomic displacements. The insets show selected displacement patterns corresponding to indicated modes. **b** Total energy from DFT as a function of the displacement amplitude for atomic displacements toward relaxed crystal structures and their projections onto soft phonon modes. **c**–**e** Relaxed crystal structures on $9\times\sqrt{3}R30°$, $7\times\sqrt{3}R30°$, and $4\times4$ unit cells from DFT. Vanadium and sulfur atoms are represented by black and yellow dots, their undistorted positions by gray shadows. Arrows represent the projections of the atomic displacements onto soft phonon modes. (Only arrows longer than 2% of the lattice constant are shown.) The contributions of different phonon modes are quantified in the figure titles. The displacements in **c**, **d** are drawn to scale, those in **e** have been magnified by a factor of three for better visibility. The primitive cell of the structure in **c**, which is in agreement with the results of ref. [21], is outlined in yellow. Dashed lines and crosses mark reflection planes and inversion centers.

small displacements. However, for larger displacements, the structures corresponding to the experimental wavevector reach by far the lowest values. These large energy gains at large displacements are inaccessible without non-linear mode–mode coupling, i.e., without the contribution of stable phonon modes (triangle marks). In the next section, we will address the non-linear regime of the distortions in a quantitative manner.

**Non-linear mode–mode coupling**. We decompose the entirety of atomic displacements of the relaxed $7\times\sqrt{3}R30°$ structure as $\mathbf{u}+\mathbf{v}$, where $\mathbf{u}$ points in the direction of the unstable transverse–acoustic phonon modes at $\mathbf{q}=\pm9/14\ \overline{\Gamma K}$ and the orthogonal complement $\mathbf{v}\perp\mathbf{u}$ combines contributions from all other phonon modes. The unstable modes account for $|\mathbf{u}|^2/|\mathbf{u}+\mathbf{v}|^2\approx69\%$ of the total displacement only. In Fig. 2b, we have already seen one-dimensional cross sections of the Born–Oppenheimer energy surface, $E(\alpha\mathbf{u})$ and $E(\beta(\mathbf{u}+\mathbf{v}))$, where $\alpha$ and $\beta$ are dimensionless scaling factors. Now, we will consider the full 2D Born–Oppenheimer surface spanned by $\mathbf{u}$ and $\mathbf{v}$. Figure 3 shows $E(x\mathbf{u}+y\mathbf{v})$, where the minimum at $x=y=1$ corresponds to the $7\times\sqrt{3}R30°$ structure and $x=y=0$ is the undistorted structure. A fourth-order polynomial fit,

$$\frac{E(x\mathbf{u}+y\mathbf{v})}{\text{meV}/\text{VS}_2}\approx-25x^2+29y^2+34x^3-99x^2y-20xy^2-12y^3$$
$$+0.1x^4+44x^3y+13x^2y^2+4.8xy^3+7.2y^4,$$

(1)

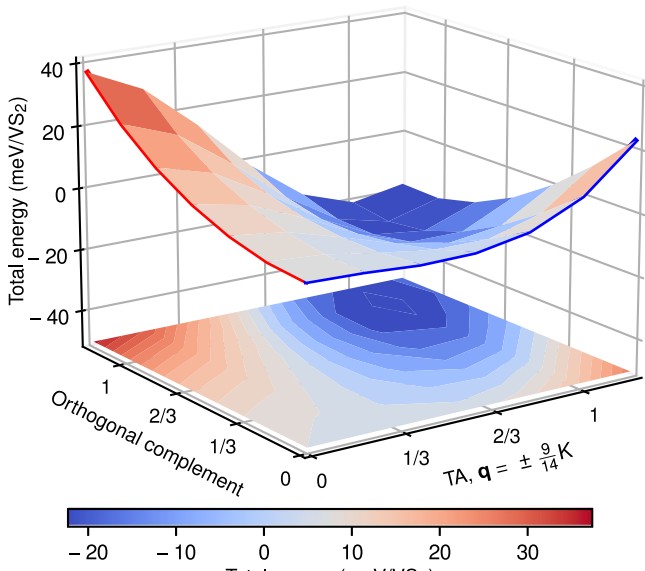

**Fig. 3 Born–Oppenheimer energy surface for the $7\times\sqrt{3}R30°$ structure of 1T-VS₂.** The axes represent the projection of the full CDW displacement onto the transverse–acoustic (TA) phonon modes at $\mathbf{q}=\pm9/14\ \overline{\Gamma K}$ and the orthogonal complement, which combines all other contributing modes. The full CDW displacement is located at the point (1, 1). The forces resulting from this energy surface are predominantly non-linear and coupled in both directions.

accurately describes the DFT Born Oppenheimer surface. Here, the first and second line give rise to linear and non-linear forces $\mathbf{F} = -\nabla\mathbf{E}$, respectively. It turns out that the non-linear part of the forces is dominated by mode–mode coupled terms[42–44] (dependent on both $x$ and $y$). The energy reduction stems largely from the $x^2y$ and $xy^2$ terms above, which correspond to a shift of the minimum of the potential-energy surface toward finite positive $y$ upon finite displacement in $x$ direction and a softening of the effective spring constant in $y$ direction for finite positive $x$, respectively. Note that within the harmonic approximation the $x^2$ ($y^2$) term lowers (raises) the energy.

The decisive role of mode–mode coupling terms $x^2y$ and $xy^2$ distinguishes 1T-VS$_2$ from systems like 2H-NbSe$_2$ or 2H-TaS$_2$, where a single mode can be employed to describe anharmonicities, and distortions along a single effective coordinate suffice to explain the relaxation pattern of the full CDW and associated energy gains (cf. Supplementary Fig. 5).

The non-linear mode–mode coupling also manifests in monolayer 1T-VTe$_2$, which is isoelectronic to monolayer 1T-VS$_2$. Monolayer 1T-VTe$_2$ in experiment realizes a $4 \times 4$ CDW[45] in contrast to monolayer 1T-VS$_2$. In line with experiment, the comparison of DFT total energies in the fully relaxed supercells (Supplementary Table 1) reveals a clear preference of the $4 \times 4$ structure in 1T-VTe$_2$. At the harmonic level, this is likely related to a shift of the lattice instabilities, especially in the transverse-acoustic branch, toward smaller wavevectors in 1T-VTe$_2$ as compared to 1T-VS$_2$ (cf. Fig. 2a and Supplementary

Fig. 6a), which can be traced back to differences in the Fermi surface (cf. Supplementary Figs. 4 and 6b–g). At the harmonic level, a CDW with $7\times \sqrt{3}R30°$ structure of monolayer 1T-VS$_2$ is not expected, as Supplementary Fig. 6a shows. The small energy gain and still appreciable distortions obtained from the relaxation of a $7\times \sqrt{3}R30°$ structure of monolayer 1T-VS$_2$ (Supplementary Table 1) despite the stability on the harmonic level suggest that non-linear mode–mode coupling is also effective, here.

**Full CDW gap in the unoccupied states**. To better understand this CDW phase, we determined the electronic structure of monolayer VS$_2$ by a combination of STS experiments and simulated d$I$/d$V$ maps based on the ab initio calculations using the $7\times \sqrt{3}R30°$ and $9\times \sqrt{3}R30°$ unit cells. STS spectra were used to locally probe the DOS of monolayer VS$_2$ at 7 K (black line) and 78.5 K (purple line), shown in Fig. 4a. Both spectra were taken with a clean Au tip in the middle of VS$_2$ islands. The most prominent feature is the gap located at about 0.175 eV, which is absent in calculations of undistorted monolayer VS$_2$[21]. At 7 K, the d$I$/d$V$ signal vanishes completely, corresponding to a full gap in the DOS. At 78.5 K, this gap is not fully open, appearing as a wide depression with a finite value at its minimum. In most other characteristic features the spectra agree qualitatively.

While the lack of energy resolution at 78.5 K certainly smears out the spectra and the gap, the reason for its absence is not immediately evident. When discussing the band structure below,

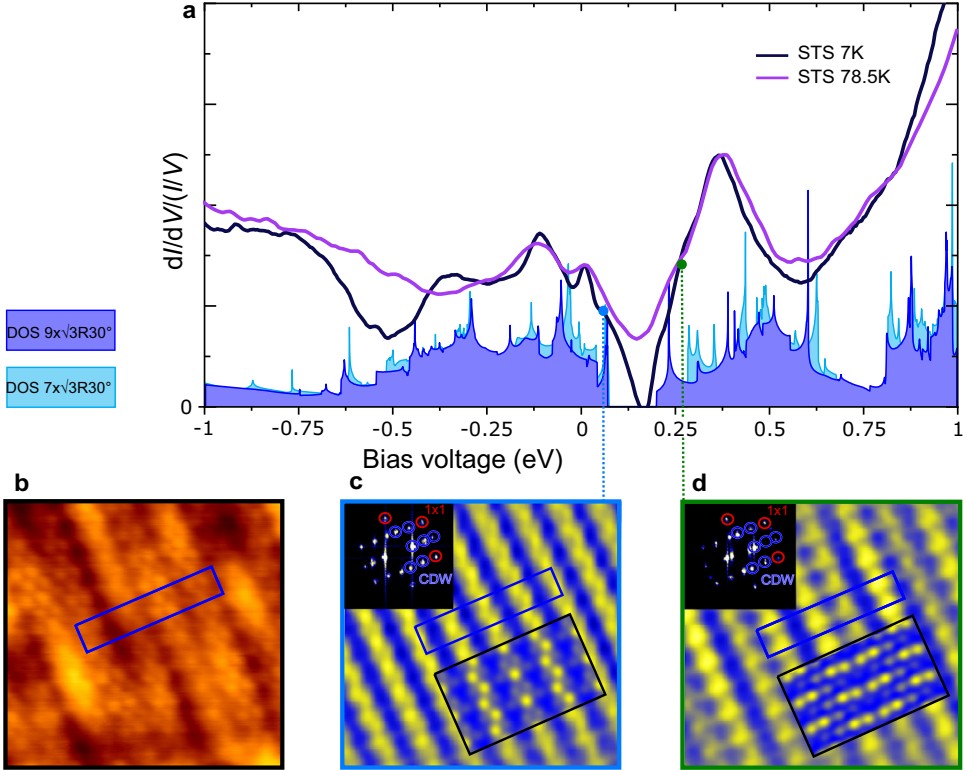

**Fig. 4 Spatially and electronically resolved CDW phase in monolayer VS$_2$. a** Scanning tunneling spectroscopy (STS) d$I$/d$V$ spectra taken with a Au tip on monolayer VS$_2$ at 78.5 K (purple) and 7 K (black). The spectra are plotted along with the DFT-calculated density of states (DOS) for the $7\times \sqrt{3}R30°$ (cyan) and $9\times \sqrt{3}R30°$ (indigo) CDW phases of monolayer 1T-VS$_2$. **b** Atomically resolved STM topograph of monolayer VS$_2$ taken at $V_t = 175$ meV. **c, d** Fourier-filtered d$I$/d$V$ conductance maps of the same region as in **b**, taken at $V_t = 75$ meV (**c**) and $V_t = 275$ meV (**d**). A linear yellow (maximum) to blue (minimum) color scale is used to depict the d$I$/d$V$ intensity. The blue box indicates the same location in **b**–**d** and corresponds to a single $9\times \sqrt{3}R30°$ unit cell of the CDW. In the same color scale, DFT-simulated d$I$/d$V$ maps below (**c**) and above (**d**) the gap of the charge density wave (CDW) are overlaid as insets. The maps show the integrated DOS from 0 to 137 meV (**c**) and from 137 to 275 meV (**d**). Additionally, the Fourier transforms of the conductance maps are shown in the upper-left corners with the $1\times 1$ (red) and CDW peaks (blue) highlighted by circles. Measurement parameters: $f = 777.7$ Hz, **a** $T = 78.5$ K, $I_t = 0.3$ nA, $V_{r.m.s.} = 6$ meV and $T = 7$ K, $I_t = 0.45$ nA, $V_{r.m.s.} = 4$ meV, **b**–**d** $T = 7$ K, $5.5 \times 5.5$ nm$^2$, $I_t = 0.3$ nA, $V_{r.m.s.} = 10$ meV.

it will be seen that the width and existence of the full gap depend on the magnitude of the lattice distortions, which may already be diminished at 78.5 K.

In the same figure, ab initio calculations for the DOS of $VS_2$, structurally relaxed in the $7\times\sqrt{3}R30°$ (cyan) or $9\times\sqrt{3}R30°$ (indigo) unit cell, are shown. Both unit cells feature quite similar structures, as expected for close-lying **q** vectors. Most striking, for both cases a full gap in the unoccupied states is predicted. They only differ in size: 0.13 eV and 0.21 eV for the $9\times\sqrt{3}R30°$ and $7\times\sqrt{3}R30°$ unit cell, respectively. The location of the gap matches the STS data. That the width of the gap in the spectrum is smaller than in DFT might stem from the ground-state calculation assumed in DFT, overestimating the vanadium atom displacement at realistic temperatures. Note also that while many of the characteristic features of calculated and measured DOS (peaks, minima) seem to agree, the experimental spectra appear to be compressed with respect to the DFT calculated DOS. This quasiparticle renormalization is indicative of strong electron–electron correlations beyond the approximations of DFT (compare Supplementary Fig. 7).

With theory and experiment largely agreeing on the electronic structure, we turn to the relation between the gap and the CDW measured on the $VS_2$ islands. For that purpose, $dI/dV$ conductance maps were taken on either side of the gap (both in the unoccupied states), in the location shown in Fig. 4b. The maps help to distinguish structural from electronic contributions, providing a close approximation of the spatial distribution of the DOS at the selected energies. As shown in Fig. 4c, d, we find two different DOS distributions on either side of the gap (see Supplementary Fig. 8 for the in-gap DOS). Both distributions are locked into the distorted lattice periodicity. They are out-of-phase, as seen in the blue unit cell drawn in the same location in Fig. 4b–d: The DOS maxima below the gap correspond to DOS minima above the gap and vice versa. This behavior is perfectly analogous to that for a CDW with a symmetric gap around the Fermi level[4]. Simulated $dI/dV$ maps derived from the DFT DOS for a $9\times\sqrt{3}R30°$ CDW are shown as an overlay in Fig. 4c, d. In Fig. 4c, the simulation reproduces both the alternating rows of single and zigzag atoms and the DOS minima between the rows. Its counterpart in Fig. 4d shows higher DOS contrast than experiment, but presents the same qualitative features. With the simulated maps based on the displacement patterns of Fig. 2c, the close agreement with experiment emphasizes the need to look beyond the harmonic approximation to understand this type of CDW.

**Band structure and Fermi-surface topology**. To deepen our understanding of the system, we calculated the spin-degenerate band structure, density of states, and Fermi surface of monolayer 1T-$VS_2$ with DFT. The results are shown in Fig. 5 and the Supplementary Movies. In the undistorted case, we find a single electronic band at the Fermi level, which strongly disperses between M and K and features a Van Hove singularity in the unoccupied states, as shown in Fig. 5a. The Fermi surface, depicted in Fig. 5b, consists of cigar-shaped electron pockets around the M points. For small distortions, partial gaps open at the Fermi level (e.g., between M and K). With increasing amplitude of the distortion, the gaps become larger and the bands are heavily reconstructed also for high energies. Only then, a full gap as observed in STS at 7 K emerges (cf. Supplementary Movie 1).

The presence of the CDW is therefore in the first place correlated with the gap between M and K, which opens already for small displacements and results in a partial gapping of the

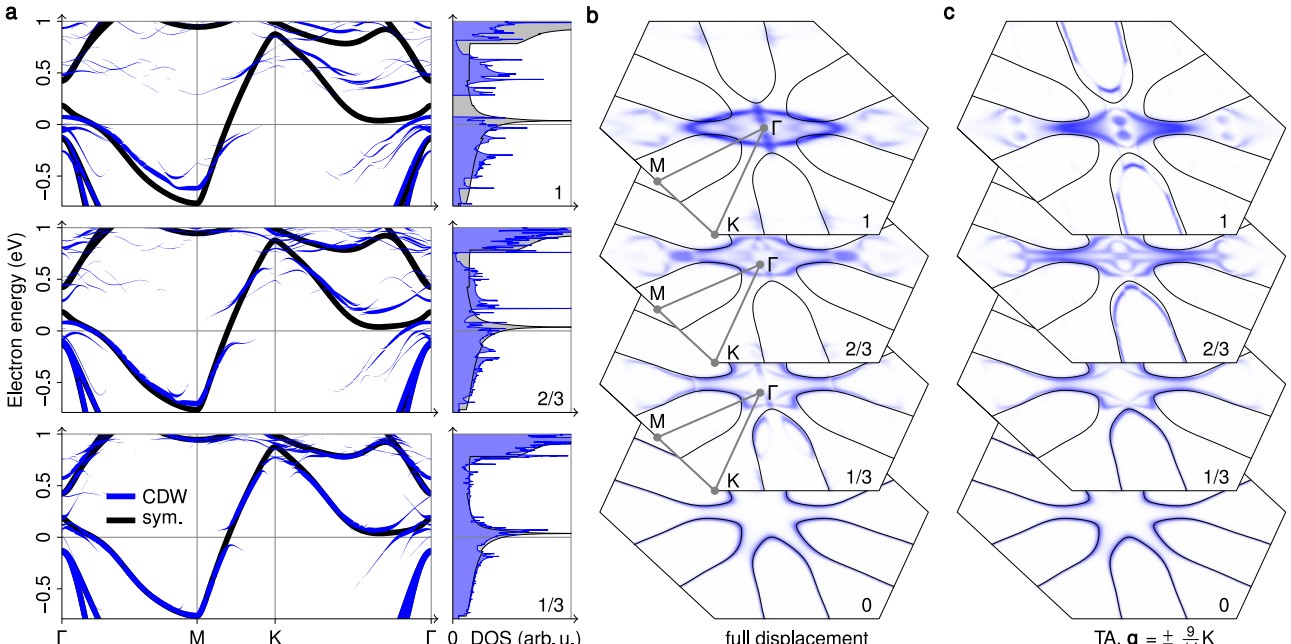

**Fig. 5 Electronic structure of monolayer 1T-VS₂ from DFT.** Data for the undistorted structure and the $7\times\sqrt{3}R30°$ CDW is shown in black and blue, respectively. The CDW data has been unfolded to the Brillouin zone of the undistorted structure. Here, the linewidth/saturation corresponds to the overlap of CDW and undistorted wave functions for the same **k** point. Analogous results for 1T-VSe₂ and the $9\times\sqrt{3}R30°$ CDW are shown in Supplementary Fig. 3. **a** Electronic band structure and density of states (DOS) for 0 %, 1/3, 2/3, and 100 % of the displacements of the relaxed CDW structure. Please note that since the CDW breaks the $C_3$ symmetry, the chosen path, indicated in **b**, does not represent the full Brillouin zone. The bands along an extended path are shown in Supplementary Fig. 9. **b**, **c** Lifshitz transition. The Fermi surface is shown for displacements toward the relaxed CDW structure (**b**) and its projection onto unstable transverse-acoustic (TA) phonon modes (**c**). The Supplementary Movies 1 and 2 show animations of the transitions in **b** and **c**, respectively (including bands, DOS, and structures).

total DOS. Presumably it is the associated gain in electronic energy that initially drives the CDW transition. Since the full gap only starts to open at 70% of the final displacements, the experimental observation of a full gap above the Fermi level is an indication that the displacements in the experiments do not fall much below the calculated ones.

At the Fermi level, there is no complete gap even at large distortion, since the downwards-dispersing bands along Γ–M are only slightly shifted downwards and remain above the Fermi level near Γ. On the other hand, the originally flat portion of the band structure between Γ and K now disperses downwards and crosses the Fermi level. The preservation of states near Γ that mask the partial gap at the Fermi level can be understood in terms of band characters and degeneracies, as shown in Supplementary Fig. 10. Altogether, the Fermi surface is reconstructed and not completely destroyed by the lattice distortion. The CDW transition is thus a metal–metal Lifshitz transition with a change in Fermi-surface topology, instead of the usual metal–insulator Peierls transition.

As shown in Fig. 5c, we cannot understand this Fermi-surface reconstruction based on a single unstable mode: The displacements expected from the harmonic approximation ($\mathbf{u}$ in Eq. (1)) only induce gaps in two-third of the cigar-shaped electron pockets (cf. Supplementary Movie 2). The other component $\mathbf{v}$ couples to segments of the Fermi surface that are not affected by $\mathbf{u}$, i.e., the remaining third of the electron pockets (cf. Supplementary Movie 3). Together, they transform the Fermi surface from multiple cigar-shaped electron pockets around the M points to the single elliptical hole pocket around Γ visible in Fig. 5b. The decomposition of the CDW contains modes at more than one wavevector $\mathbf{q}$, so several approximate Fermi-surface nesting conditions and electron–phonon coupling matrix elements play a role (Supplementary Fig. 4b, c, e, f), enabling the CDW to affect distinct parts of the Fermi surface.

**Magnetic properties of monolayer VS₂.** Prompted by the prediction of ferromagnetism for monolayer 1T-VS₂ in its $\mathbf{q} = 2/3\ \overline{\Gamma K}$ CDW phase[21], we also examined the magnetic properties of VS₂, by means of X-ray magnetic circular dichroism (XMCD). The monolayer VS₂ samples were grown in situ and investigated with STM beforehand to make sure that the same phase and decent coverage were obtained. A STM topograph of the sample investigated by XMCD is shown in Supplementary Fig. 11. The blue curve in Fig. 6a represents the X-ray absorption spectrum averaged over both helicities and external field directions. The overall line shape is very similar to previous bulk crystal measurements[23] and clearly fits to a $3d^1$ configuration[46]. The red signal in Fig. 6a is the XMCD magnified by a factor of 10, where no signal above the noise level is visible. This implies that the total magnetization vanishes. Sum rule analysis would yield an upper

bound of $0.02\mu_B$ per vanadium atom. Since it cannot be strictly applied to the case of the $V_{2,3}$ edges[47], this analysis yields only a zero-order estimate of the upper bound, but we can safely conclude that neither ferromagnetic nor paramagnetic behavior is present in this system.

We investigated magnetic order in monolayer VS₂ using spin-polarized DFT. We were able to stabilize both ferromagnetic and SDW structures within the $7 \times \sqrt{3}R30°$ unit cell. In fact, magnetically ordered CDW phases are preferred over nonmagnetic CDW phases by energies of the order of 1 meV per VS₂ unit. Figure 6b shows the most favorable SDW pattern in the $7 \times \sqrt{3}R30°$ CDW phase. The magnetic moments on vanadium reach $\pm0.18\mu_B$, those on sulfur only $\pm0.01\mu_B$ and are thus not shown. While the CDW alone reduces the total energy to $-22.7$ meV per VS₂ unit with respect to the symmetric structure, the SDW lowers this value by another 1.5 to $-24.2$ meV. Interestingly, without the CDW, a similar SDW with larger local moments of up to $\pm0.51\mu_B$ (shown in Fig. 6c) leads to an energy reduction of 7.1 meV. As already suggested by previous calculations of ferromagnetism in the $9 \times \sqrt{3}R30°$ structure of 1T-VS₂[21], there is a competition between the lattice distortion and the formation of local moments. Although a full account of magnetism needs to go beyond the DFT level, in view of the good agreement between our ab initio results and the experimental STS and XMCD data, the formation of coupled CDW–SDW state in 1T-VS₂ is plausible. This presumption is further supported by comparison of the calculated DOS in the CDW–SDW state to the STS shown in Supplementary Fig. 12: the SDW formation on top of the CDW leads to a reduction of the gap size and the DOS of the coupled CDW–SDW is even in better agreement with the experiment than the non-spin-polarized CDW DOS.

## Discussion

Both the electronic and magnetic results for VS₂ shed some light on the properties of the isoelectronic compound VSe₂, which displays a CDW of the same periodicity[3,38,48]. Our calculations strongly suggest that also for this system non-linear effects are relevant and that a full gap opens in the unoccupied states (compare Supplementary Fig. 3). A full gap at the Fermi level, as proposed for 1T-VSe₂[6,38,48,49], would be unlikely to intrinsically occur for a CDW with the observed wavevector. The strong similarity between our calculations and experimental data, especially for those VSe₂ systems where only the $7 \times \sqrt{3}R30°$ CDW is observed[36,37], lends credence to our analysis (compare Supplementary Fig. 13). It is possible that the simultaneous occurrence of a $4 \times 1$ CDW[3,6], perhaps due to substrate-induced strain[50], causes an additional gap opening near the Fermi level as a result of the interplay between the CDWs. In any case, similar to VS₂, the presence of a SDW coupled to the $7 \times \sqrt{3}R30°$ CDW

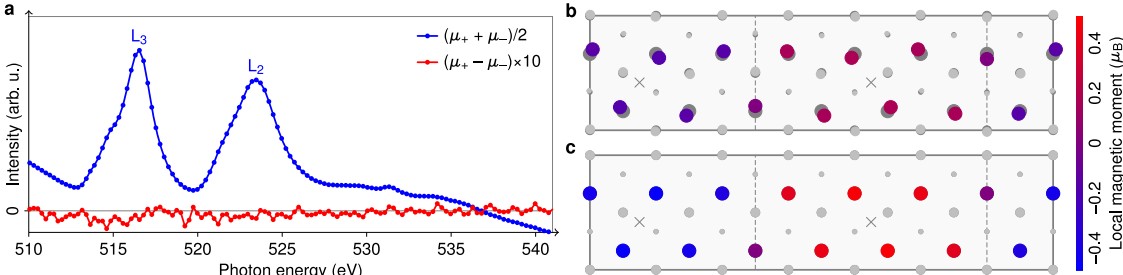

**Fig. 6 Magnetic properties of monolayer 1T-VS₂. a** Plotted in blue is the X-ray absorption signal averaged over both helicities and directions of the *B* field. The corresponding XMCD is shown in red. All measurements have been conducted in *B* fields of ±9 T and at a temperature of 7 K. **b**, **c** Possible SDW pattern with (**b**) and without CDW (**c**).

could explain the absence of net magnetization in XMCD experiments[37,38,48]. Spin-polarized STM or XMLD might be able to detect the magnetic ground state for both $VS_2$ and $VSe_2$.

In conclusion, $VS_2$ defies the common phenomenology of CDW formation, as the complete CDW gap occurs above the Fermi level, there is giant non-linear longitudinal–transverse mode–mode coupling, and the CDW formation is accompanied by a change of the Fermi-surface topology. The unconventional CDW appears to host further electronic correlations as signalled by the quasiparticle renormalization and magnetic-moment formation. In this respect, it is reminiscent of correlated phases in superlattice structures such as Star of David phases[7], moiré superlattices[51], and doped cuprate superconductors[52]. In the latter class, lattice anharmonicities are central to boosting superconductivity under THz optical driving[44]. The case of $VS_2$ presents new terrain: A metal–metal Lifshitz transition from non-linear electron–lattice effects in the strong-coupling regime is intertwined with electronic correlations. We note that the full gap in the DOS, situated within 0.2 eV from the Fermi level, opens up the possibility of inducing a metal–insulator transition upon mild gating or doping (e.g., with Li). Finally, we are convinced that the excellent agreement of experiment and theory for the unconventional CDW of monolayer $VS_2$ with the full gap in the unoccupied states provides a paradigmatic case study of strong-coupling CDWs in general.

## Methods

The Ir(111) crystal is cleaned by grazing incidence 1.5 keV $Ar^+$ ion exposure and flash annealing to 1500 K. A closed monolayer of single-crystalline Gr on Ir(111) is grown by room temperature exposure of Ir(111) to ethylene until saturation, subsequent annealing to 1300 K, followed by exposure to 200 L ethylene at 1300 K[53].

The synthesis of vanadium sulfides on Gr/Ir(111) is based on a two-step MBE approach introduced in detail in ref. [34] for $MoS_2$. In the first step, the sample is held at room temperature and vanadium is evaporated at a rate of $F_V = 2.5 \times 10^{16}$ atoms/($m^2$s) into a sulfur background pressure of $P_S^g = 1 \times 10^{-8}$ mbar built up by thermal decomposition of pyrite inside a Knudsen cell. This results in dendritic TMDC islands of poor epitaxy. To make the islands larger and more compact, the sample is flashed in a sulfur background to 600 K.

The $VS_2$ layers were analyzed by STM, STS, and low-energy electron diffraction (LEED) inside a variable temperature (30–700 K) ultrahigh vacuum apparatus and a low-temperature STM operating at 7 and 78.5 K. The software WSXM[54] was used for STM data processing. XMCD measurements have been conducted at the beamline ID32 of the European Synchrotron Radiation Facility (ESRF) in Grenoble, France. The $VS_2$ samples were grown in situ inside the preparation chamber and checked with LEED and STM before X-ray absorption spectroscopy measurements. To be surface sensitive, the measurements were conducted in the total-electron-yield mode under normal incidence. The measurement temperature was 7 K and fields of 9 T were used. The spectra were recorded at the $L_{3,2}$ edges, i.e., using the dipole allowed transition from $2p$ states into the $3d$ shell potentially generating magnetism.

All DFT and DFPT calculations were performed using QUANTUM ESPRESSO[55,56]. We apply the PBE functional[57,58] and norm-conserving pseudopotentials from the PSEUDODOJO table[59,60]. In the undistorted case, uniform meshes (including $\Gamma$) of $12 \times 12$ **q** and $24 \times 24$ **k** points are combined with a Fermi–Dirac smearing of 300 K. For a fixed unit-cell height of 15 Å, minimizing forces and in-plane pressure to below $1 \times 10^{-5}$ Ry/Bohr and 0.1 kbar yields a lattice constant of 3.18 Å and a layer height (vertical sulfur–sulfur distance) of 2.93 Å. For the superstructure calculations, appropriate **k**-point meshes of similar density are chosen, except for the precise total energies quoted in the section about magnetism and in Supplementary Table 1, which required four times as dense meshes. The average lattice constant of superstructures is kept fixed at the value of the symmetric structure. Fourier interpolation to higher **k** resolutions ($1000 \times 1000$ for calculations of the DOS) and the unfolding of electronic states is based on localized representations generated with WANNIER90[61]. For the visualization of the unfolded Fermi surfaces, a Fermi–Dirac broadening of 10 meV is used.

## Data availability

All the data and methods are present in the main text and the supplementary materials. Any other relevant data are available from the authors upon reasonable request.

## Code availability

Codes used in this work are available from the authors upon reasonable request.

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

## Acknowledgements

This work was funded by the Deutsche Forschungsgemeinschaft (DFG, German Research foundation) through CRC 1238 (project no. 277146847, subprojects A01 and B06). J.B., A.S., and T.Weh. acknowledge financial support by the DFG through EXC 2077, GRK 2247, and SPP 2244. T.Weh. acknowledges support via the European Graphene Flagship Core3 Project (grant agreement 881603). J.B., A.S., and T.Weh. acknowledge computational resources of the North-German Supercomputing Alliance (HLRN). E.v.L. is supported by the Central Research Development Fund of the University of Bremen. L.M.A. acknowledges financial support from CAPES (project no. 9469/13-3).

## Author contributions

T.Weh. and T.M. conceived this work and designed the research strategy. J.H. and discovered the CDW and developed, with the assistance of T.Wek., the growth method. C.v.E. and E.P. conducted and analyzed the STS experiments. J.B., A.S., and G.S., with support from E.v.L. and T.Weh., performed ab initio calculations. F.H. and S.K., with support from N.R., K.O., L.A., N.B., K.K., and H.W., performed and analyzed the XMCD experiment. The results were discussed by all authors. C.v.E., J.B., J.H., E.v.L., S.K., T.Weh., and T.M. wrote the manuscript with input from all authors. The first three authors have comparable contributions to this work.

## Funding

## Competing interests

The authors declare no competing interests.
