## [Peer Review File · Nature Communications]

A full gap above the Fermi level: the charge density wave of monolayer VS₂REVIEWER COMMENTS

Reviewer #1 (Remarks to the Author):

The authors present a combined STM/STS and DFT study of the CDW in monolayer VS₂. They draw analogy to the more frequently studied case of VSe₂, which exhibits a similar CDW for the monolayer. The large CDW unit cell agrees with the imaginary phonon modes for the transverse phonons. The longitudinal phonons also predict an instability but would give a 4x4 periodicity, which, however, is not observed experimentally. If possible it would be interesting to discuss why the longitudinal phonon instability is not observed, especially in comparison to the other two V-dichalcogenides, where VSe₂ shows a 4x4 for the bulk but not for the monolayer, and VTe₂ has recently been demonstrated to exhibit a 4x4 in the monolayer. This seems to imply that sometimes the longitudinal phonon modes determine the CDW instability and in other cases the transverse phonon modes. What determines this selection? In the calculations what is the relaxed structure for a 4x4 unit cell? Is the relaxed structure of the 4x4 unit cell higher than that for the transverse unit cell? If not, why is the 4x4 not observed? It seems that longitudinal atom displacements are important to explain the electronic structure variation and partial gap opening at the Fermi-level (it has been shown that VTe₂ opens a partial gap at the BZ boundary and the calculations in this manuscript show that the longitudinal displacements are necessary to open a partial gap). Is it a coincidence that the longitudinal phonon wave-vector is the one that is close to a Peierls' like nesting condition and that the opening of the gap at the Fermi-level (at the BZ-boundary) is where the nesting is observed? Does a Peierls' condition always imply a gap over the entire BZ? Or could a partial gap consistent with a Peierls' condition? So, could one claim that the longitudinal mode is Peierls' like and the transverse mode is pure phonon instability. This all may be speculation, but in how far does the current manuscript answer any of these questions? Maybe some more discussion is needed. The Peierls' model is generally believed to be an oversimplification in 2D materials, and a phonon instability seems to describe most CDW transitions in TMDs well, so what is the transformative contribution in the current work? Is it just the contribution of anharmonicity that is required to explain the STS data? I am not sold that this is a breakthrough in the understanding of CDW in TMDs, but it is a nice contribution to the understanding of CDW monolayers in V-dichalcogenides and thus may be published with some more discussion as outlined in this report. From STS a 'full' band gap is observed above the Fermi-level and this can be reproduced from DFT calculations using the CDW unit cell and relaxing the atom positions. This optimized structure is lower in energy than the 1x1 1T structure, but it also contains horizontal atom displacements in addition to the lateral atom displacements of the transverse phonon modes. Thus, significant anharmonicities are implied. These anharmonicities (vertical atom displacements) are required to open the observed gap and also a partial gap at the Fermi-level around the BZ boundary, which has been observed for VSe₂ in some ARPES studies. Thus, there seems to be good agreement between theory and the STS and dI/dV mapping which supports the theoretical studies. It would be reassuring, though, if the authors could also demonstrate the agreement with ARPES studies for VSe₂, beyond a partial opening of a gap at the Fermi-level at BZ boundaries. ARPES measurements for VSe₂ exhibits fairly small changes during the transition from the 'normal' to the CDW-phase (apart from a (partial) opening of a gap). I am wondering if this is consistent with the calculations shown in the manuscript. It is difficult to judge from the presented calculations. To validate the accuracy of the calculations, a comparison with literature data of the 'Fermi' -surfaces (constant energy surfaces) for different binding energies should be shown (averaging over the different domains would be required). If it does not reproduce the reported experimental data, the theoretical approach may need to be reconsidered. Measuring the k-resolved electronic structure in ARPES can give much more detailed and reliable information and thus it would be nice to have ARPES data for VS₂, also. The authors claim that the synthesis of VS₂ is challenging and their growth they obtain VS₂ with different orientation and thus this prevents ARPES studies. However, is this just a consequence of the chosen substrate (graphene/Ir(111))? And the growth conditions required by the substrate. Have the authors tried to grow on the more conventional graphene/SiC material, which may allow to grow directly at elevated temperatures. It should probably be explored if this gives better samples that could also be investigated by ARPES.

Reviewer #2 (Remarks to the Author):

The authors investigate experimentally and theoretically the occurrence of charge density wave formation in single layer VS₂. The results are potentially interesting but there are a substantial number of points that should be clarified in order for this paper to be publication quality. The present version is not convincing and not suitable for publication in nature communications. The author should try to submit a revised version addressing the following critical points:

1) The use of the term "anharmonic" is misleading in all the paper. The authors use this term to state that, as the charge density wave phase involves large displacements with respect to the undistorted structure, there is a substantial admixture of transverse and longitudinal modes. However, they never calculate any anharmonic effect. Their free energy is harmonic (as far as I can understand from the paper, as there are no details) and their calculations in the CDW and non CDW phases are either total energy calculation or harmonic calculations. The free energy difference would be enormously affected by anharmonicity (calculation not included in this paper) and if anharmonicity were included the results would change substantially.

What the authors mean is that as the CDW involves large atomic displacements, the CDW phase cannot be seen as a linear (small) perturbation of the undistorted phase and structural optimization must be performed. This is, by the way, not surprising as it is similar in other dichalcogenides. and explains why the Peierls picture breaks down. The large perturbation and the symmetry reduction of course allows for coupling of different modes. Thus, the authors should replace the term "anharmonic" (clearly misleading) with statements related to the non linear regime of the distortion.

2) The authors mention that "the total free energy is reduced by 24 meV per VS₂ unit". It is not clear what the authors mean here. Is the total free energy the harmonic free energy including both the phonon (vibrational) contributions and the smearing (Fermi-Dirac) contribution ? Are the author conscious that the statement that the free energy gain of 24 meV / VS₂ unit means an enormous energy gain in the supercell, order of magnitudes larger than what expected for a 300 K T_{CDW} (and much larger of what found in other single layer TMDs) ? This should be rationalized a bit better, double checked and carefully compared with other calculations performed on 2D dichalcogenides. What is the origin of this gigantic energy gain at the PBE level ? The authors should give a clear motivation of why this dichalcogenides is different from the others despite the practically identical electronic structure. Furthermore, the authors should also quote the total energy difference, so that we can distinguish if the difference arises from the vibrational part or not.

Finally, I am a bit confused by the harmonic free energy calculation for the undistorted phase. The harmonic free energy is well defined only for positive phonon frequencies as the resummation of the partition function requires the argument of the geometric serie $\sum_n \exp(-\beta \hbar \omega_n)$

to be smaller than one, otherwise the partition function diverges. How did the authors fix this problem in calculating the free energy difference between the two phases ? Could it be that this is at the origin of their large free energy difference ?

3) The interesting XMCD data at the Vanadium L_{2,3} edges exclude the occurrence of a ferromagnetic ground state, as correctly noted by the authors. However, this does not exclude an antiferromagnetic ground state.

The authors indeed found that the AF ground state is more stable in the CDW phases, however they did not characterize it (despite being the ground state!), mostly because they believe that calculations with other exchange and correlation kernels are needed. My feeling is that if the AF state is stable even

at the PBE level, there are strong possibility that this is the real ground state.

Thus, the authors should (i) calculate the energy gain of the AF-CDW state with respect to the (magnetic) ground state of the undistorted structure, (ii) calculate and quote the portion of the magnetic energy gain with respect to the structural energy gain, (iii) calculate the density of states of the CDW magnetic structure.

How does it compare with the non magnetic one and with experiments ?

4) The graphical quality of the unfolded bands in blue in Fig. 4 (a) is very poor (the same for the pictures in the supplemental materials) and does not seem to match the one of Fig. 4(c). For example, the Fermi surface

in 4(c) exhibit a Fermi level crossing at approximately 1/2 of Gamma-K. This crossing is invisible in Fig. 4 (a). To me this is because the supercell breaks C₃ and C₆ symmetries and the different Gamma-K

directions of the undistorted structure Brillouin zone are not anymore equivalent when the bands are unfolded. In Fig. 4 (a) the authors only pick one of these directions. Can the author improve the picture

4 (a) so that to include also the other Gamma-K direction with the elongated part of the cigar like Fermi surface crossing E_f ?

5) Finally a crucial point. The authors underlined the opening of a band gap by the distortion in the unoccupied region. However, they give no explanation for it. Why the gap opens not at E_f but somewhere else ? This is a crucial point that the authors cannot avoid explaining.

Reviewer #3 (Remarks to the Author):

The physics driving the charge density wave (CDW) transition in solid-state crystals, and the role of Fermi surface nesting, have been longstanding subjects of debate within the solid-state community. The authors of this new manuscript make an important contribution to this discussion. Using a combination of scanning tunnelling microscopy/spectroscopy (STM/S), ab-initio calculation, and x-ray magnetic circular dichroism (XMCD), they argue compellingly that the CDW state in single layer VS₂/graphene/Ir(111) is driven by strong electron-phonon coupling, with little or no role for Fermi surface nesting. The paper presents many important new insights into the scope of the CDW phenomenon, the fundamental physics that underlie it, and its possible interplay with magnetism. The quality of the data is very good, and the calculations give good agreement with the experimental results. The findings are novel and will be of keen interest to the community, and are likely to stimulate future research in more than one direction:

1. "unconventional" CDW states associated with strong electron-phonon coupling and gaps opening away from the Fermi level—this latter being, to my knowledge, a topic which has not been widely recognised or understood before now;

2. interaction between CDWs and magnetism;
3. metal-metal Lifshitz transitions generated by electron-phonon-driven CDW transitions—an intriguing idea that only becomes possible once one moves beyond the Peierls' view.

All three of these directions are “hot topics” within solid-state physics. Since there is no reason to think that the phenomena explored in the manuscript are limited to the particular material system treated in the study, or even to the broader category of two-dimensional materials, the work will be relevant to specialists spanning a wide spectrum of the solid-state research community.

I might also note that high-quality VS₂ is notoriously difficult to prepare, and the results presented in the manuscript point to very high sample quality. The authors have developed their growth method—using pyrite as a sulphur source—for some years, and have successfully realised a variety of single-layer transition-metal dichalcogenides. However, the results presented here are, to my mind, particularly persuasive of the potential of the method for producing high-quality sulphide samples. Such results might stimulate larger-scale adoption of this method among other labs doing similar work.

Overall, I feel that the manuscript is appropriate to Nature Communications, on the basis of its

novelty
impact
conceptual advances
potential interest to readership
important advances of significance to specialists within each field.

I do feel that there are some issues, however, that should be addressed in the manuscript before it is accepted for publication.

Sample stoichiometry:

Previously published literature indicates that stoichiometric VS₂ can easily desulphidise into multiple S-deficient forms, both in the bulk and in the single-layer. Considering that STM does not give direct information about stoichiometry, some discussion of this issue is warranted. The authors convincingly defend their claim that the striped superstructure is the same stoichiometry as the non-striped structure. However, they do not (unless I am overlooking something) explicitly exclude the possibility that the stoichiometry of both the striped and non-striped phases could be something other than 1:2 V:S. (Note also the assertion at the top of p. 15 that STM was used to check the stoichiometry of their samples before XMCD measurements. This cannot be, strictly speaking, correct.) As a matter of fact, I do believe that the stoichiometry is 1:2, as the authors say, since the structure is hexagonal and highly ordered, with no visible defects in the atomically resolved data or di/dv mapping, and since agreement with theory is good. But I think this important issue should be discussed explicitly somewhere in the manuscript or the supplement.

Discussion of gap:

Naïvely speaking, it seems surprising that the CDW gap in Fig. 3 is already closing at 78.5K, considering that the CDW state persists to >300K. Can the authors explain this?

Discussion of V atom displacement:

The displacement of the V atoms from equilibrium is estimated to be 8%, on the basis of theory. The authors note (p. 10) that this might be an overestimate. In any case, 8% strikes me as huge. Can the authors comment on this? How does this compare to CDW distortions in other material systems? How sensitively dependent are calculated densities of states and electronic dispersions upon this value? This seems like an important issue that deserves some discussion, even if only in the supplement.

Caption to Fig. 2:

I am not sure to what “colored stripes in the background” refers. Are these the darker and lighter yellow regions?

Figs. 1, 3:

I do not see the “indigo box” that is referenced in the caption to Fig. 3, nor do I see indigo circles in the Fourier transforms. I do see grey boxes and grey circles—is it these to which the caption refers? In general, the grey is somewhat difficult to see, both in Figs. 1 & 3. The grey arrows in the insets (Fourier transforms) in panels (b) and (c) are particularly difficult to see.

P. 6:

The text reads:

“For the DFT calculations below, the experimental wave pattern must be approximated by a commensurate supercell. Two close approximations are overlaid on the atomic resolution images in Fig. 1(b, c). The gray boxes in Fig. 1(b) and (c) indicate $9 \times 3 \times 3$ and $7 \times 3 \times 3$ structure with periodicities $(2.25 \pm 0.02)a_{VS2}$ and $(2.33 \pm 0.02)a_{VS2}$, respectively.”

My understanding is that the observed superstructure is neither $9 \times 3 \times 3$ nor $7 \times 3 \times 3$, but is rather an incommensurate structure falling between these two, and that the commensurate structures are introduced for the sake of computational tractability. Is that correct? However, in Fig. 1, the $9 \times 3 \times 3$ structure is superimposed on the data acquired at 7K in panel (b), whereas the $7 \times 3 \times 3$ structure is superimposed on the 300K data in panel (c)—which would seem to suggest that the two commensurate structures are temperature-dependent phases. Perhaps the authors could change the presentation of the structural models, to avoid confusion.

Author reply to reviewer reports on “A full gap above the Fermi level: the charge density wave of monolayer VS₂”

Camiel van Efferen, Jan Berges, Joshua Hall, Erik van Loon, Stefan Kraus, Arne Schobert, Tobias Wekking, Felix Huttmann, Eline Plaar, Nico Rothenbach, Katharina Ollefs, Lucas Machado Arruda, Nick Brookes, Gunnar Schönhoff, Kurt Kummer, Heiko Wende, Tim Wehling, Thomas Michely

We would like to thank the reviewers for their time and effort spent on reviewing our manuscript. The reviewers raised many important questions that we will answer in detail below. Here, the most substantive concerns relate to the importance of our work to the field of charge density wave physics, to the adequacy of our theoretical modelling, and to possible terminological ambiguities. We are confident that the revised version of our manuscript (and the Supplementary Information) adequately addresses this criticism and is now suitable for publication in Nature Communications.

From the reports of the reviewers we learned that in the first version of the manuscript, the heart of our work was not as clearly recognizable as intended. Hence, when rewriting the manuscript, rather than on the discussion about a Peierls vs strong-coupling mechanism behind the charge density wave, we now focus more on the importance of non-linear mode–mode coupling and associated changes of the Fermi-surface topology for the structural phase transition. We also emphasize this in the discussion and abstract.

Moreover, we adapted the structure and format of our manuscript to the guidelines of Nature Communications, and we added detail where appropriate within the more relaxed length restriction of this journal. Finally, in view of the reviewers’ comments regarding the scope of our work, we would like to emphasize its collaborative character: With the aim to better understand the material of study, we address precisely those questions where both experimental and theoretical fingerprints are available.

Yours sincerely,

The authors

Reply to Reviewer 1

The authors present a combined STM/STS and DFT study of the CDW in monolayer VS₂. They draw analogy to the more frequently studied case of VSe₂, which exhibits a similar CDW for the monolayer. The large CDW unit cell agrees with the imaginary phonon modes for the transverse phonons. The longitudinal phonons also predict an instability but would give a 4 × 4 periodicity, which, however, is not observed experimentally. If possible it would be interesting to discuss why the longitudinal phonon instability is not observed, especially in comparison to the other two V-dichalcogenides, where VSe₂ shows a 4 × 4 for the bulk but not for the monolayer, and VTe₂ has

recently been demonstrated to exhibit a 4×4 in the monolayer. This seems to imply that sometimes the longitudinal phonon modes determine the CDW instability and in other cases the transverse phonon modes. What determines this selection? In the calculations what is the relaxed structure for a 4×4 unit cell? Is the relaxed structure of the 4×4 unit cell higher than that for the transverse unit cell? If not, why is the 4×4 not observed?

The authors thank the reviewer for this important point on the interplay of different charge density wave (CDW) instabilities. To address this question, we performed additional DFT calculations to determine the total energy of the fully relaxed and partially distorted structures, extended Fig. 2 (cf. panels **b** and **e**), and added Fig. 3 to the revised version of the manuscript. These new data and new accompanying discussions show that non-linear mode–mode coupling is responsible for the preference of the transversal CDW at $\mathbf{q} \approx 9/14 \overline{\Gamma\overline{K}} \approx 2/3 \overline{\Gamma\overline{K}}$ over the 4×4 structure.

We also highly appreciate the suggestion to compare to the CDWs in other V-based dichalcogenides. In our view, the 4×4 in 1T-VTe₂ is particularly suitable for a comparison, since it is observed in the monolayer as well. We therefore added a comparison of 1T-VS₂ and 1T-VTe₂ to the main text, the details of which are discussed in the Supplementary Information (SI). There we show that DFT indeed reveals in VTe₂ a preference of the 4×4 CDW structure, i.e., it has a lower total energy than the other CDW (cf. Table S1 in the revised SI). For a more detailed comparison, we repeated the fluctuation-diagnostics treatment of 1T-VS₂ (Fig. S4 in the SI) for the case of 1T-VTe₂, as shown in the new Fig. S6 in the SI. It shows that the replacement of S by Te changes Fermi-surface topology, which likely contributes to the difference in preferred CDW structures.

Manuscript and SI amended.

It seems that longitudinal atom displacements are important to explain the electronic structure variation and partial gap opening at the Fermi-level (it has been shown that VTe₂ opens a partial gap at the BZ boundary and the calculations in this manuscript show that the longitudinal displacements are necessary to open a partial gap). Is it a coincidence that the longitudinal phonon wave-vector is the one that is close to a Peierls' like nesting condition and that the opening of the gap at the Fermi-level (at the BZ-boundary) is where the nesting is observed? Does a Peierls' condition always imply a gap over the entire BZ? Or could a partial gap consistent with a Peierls' condition? So, could one claim that the longitudinal mode is Peierls' like and the transverse mode is pure phonon instability. This all may be speculation, but in how far does the current manuscript answer any of these questions? Maybe some more discussion is needed.

The authors thank the reviewer for addressing the important aspect on how the admixture of longitudinal modes to the CDW affects the electronic structure and its relation to the Peierls model. There are different longitudinal modes involved in the discussion: the leading instability on the harmonic level at $\mathbf{q} = 1/2 \overline{\Gamma\overline{M}}$ (not admixed to the experimentally observed CDW) and the longitudinal mode at $\mathbf{q} \approx 4/3 \overline{\Gamma\overline{K}}$ (involved in the observed CDW). We revised Fig. 2 of the main text to better visualize the different longitudinal and transverse modes contributing to the observed CDW and to avoid misunderstandings.

The longitudinal component of the experimentally observed CDW (i.e., at $\mathbf{q} \approx 4/3 \overline{\Gamma\overline{K}}$) is

not related to any nesting or Peierls physics (cf. Fig. S4e, f in the SI) in contrast to the longitudinal phonon at $\mathbf{q} = 1/2\overline{\Gamma M}$ (cf. Fig. S4a in the SI). We added a statement to the main text to explicate the non-Peierls nature of the $\mathbf{q} \approx 4/3\overline{\Gamma K}$ admixture to the CDW and to explain that it is the combination of multiple approximate nesting scenarios (Fig. S4b, c, e, f) that leads to the partial gap.

Manuscript amended.

The Peierls' model is generally believed to be an oversimplification in 2D materials, and a phonon instability seems to describe most CDW transitions in TMDs well, so what is the transformative contribution in the current work? Is it just the contribution of anharmonicity that is required to explain the STS data? I am not sold that this is a breakthrough in the understanding of CDW in TMDs, but it is a nice contribution to the understanding of CDW monolayers in V-dichalcogenides and thus may be published with some more discussion as outlined in this report.

The authors thank the reviewer for this point, which indeed helped us to state more clearly the central findings of our paper:

1. Rather than the strong-coupling nature of the CDW, which is ubiquitous in TMDCs and has been extensively discussed elsewhere, the first crucial aspect in our work is that the CDW is governed by non-linear mode–mode coupling. Even very elaborate treatments of strong-coupling CDWs¹ usually consider a single effective phonon mode as in the Peierls picture. Without mode–mode coupling, we would still have a strong-coupling CDW, albeit with much smaller displacements and energy reduction. Such effects are certainly not limited to monolayers 1T-VX₂, but sparsely appreciated in the literature.
2. Regardless of the strong- or weak-coupling nature of the CDW, the analysis of CDW-induced changes in the electronic structures has often focused on spectral changes at or below the Fermi level. Our work goes beyond the standard phenomenology: we reveal a full CDW gap above the Fermi level and a CDW induced metal–metal transition with associated unusual changes of the Fermi-surface topology.

To make these points clear, we rewrote the abstract of the paper, and we updated the discussion on non-linear mode–mode coupling and on the changes in electronic structure as well as the concluding paragraphs of the manuscript.

Manuscript amended.

From STS a 'full' band gap is observed above the Fermi-level and this can be reproduced from DFT calculations using the CDW unit cell and relaxing the atom positions. This optimized structure is lower in energy than the 1×1 IT structure, but it also contains horizontal atom displacements in addition to the lateral atom displacements of the transverse phonon modes. Thus, significant anharmonicities are implied. These anharmonicities (vertical atom displacements) are required to open the observed gap and also a partial gap at the Fermi-level around the BZ boundary, which has been observed for VSe₂ in some ARPES studies. Thus, there seems to be good agreement between theory and the STS and dI/dV mapping which supports the theoretical studies. It would be reassuring, though, if the authors could also demonstrate the agreement with ARPES studies for VSe₂, beyond a partial opening of a gap at the Fermi-level at BZ boundaries. ARPES measurements for VSe₂

Figure 1: Fermi surface of $7 \times \sqrt{3}R30^\circ$ CDW of monolayer 1T-VSe₂ averaged over regions of different CDW orientations and comparison to experiment. **a** Fermi surface reprinted with permission from Ref. 2 © 2018 American Physical Society. **b** Symmetrized Fermi surface calculated in this work. **c** Energy isolines reprinted with permission from Ref. 3 © 2018 American Chemical Society. **d** Symmetrized energy isolines calculated in this work.

exhibits fairly small changes during the transition from the ‘normal’ to the CDW-phase (apart from a (partial) opening of a gap). I am wondering if this is consistent with the calculations shown in the manuscript. It is difficult to judge from the presented calculations. To validate the accuracy of the calculations, a comparison with literature data of the ‘Fermi’-surfaces (constant energy surfaces) for different binding energies should be shown (averaging over the different domains would be required). If it does not reproduce the reported experimental data, the theoretical approach may need to be reconsidered.

The authors acknowledge the suggestion of the reviewer to extend the comparison of our calculations for 1T-VSe₂, shown in the SI, to existing ARPES data for that material. In Fig. 1, we show constant-energy contours of literature VSe₂ ARPES data alongside our calculated electronic structure of 1T-VSe₂ in the $7 \times \sqrt{3}R30^\circ$ CDW phase. Our calculations are now shown averaged over the three different domains of the CDW with respect to the lattice. In the manuscript, in the first paragraph of the discussion, where the similarities and differences between both materials are reviewed, the agreement of our calculations with existing ARPES data for VSe₂ is emphasized for those cases where only the $7 \times \sqrt{3}R30^\circ$ transition occurred. In addition, Fig. 1 is added to the SI, together with a detailed discussion.

Manuscript and SI amended.

Measuring the k-resolved electronic structure in ARPES can give much more detailed and reliable information and thus it would be nice to have ARPES data for VS₂, also. The authors claim that the synthesis of VS₂ is challenging and their growth they obtain VS₂ with different orientation and thus this prevents ARPES studies. However, is this just a consequence of the chosen substrate (graphene/Ir(111))? And the growth conditions required by the substrate. Have the authors tried to grow on the more conventional graphene/SiC material, which may allow to grow directly at elevated temperatures. It should probably be explored if this gives better samples that could also be investigated by ARPES.

The authors acknowledge the suggestion of the reviewer. Indeed, it would be desirable to conduct ARPES experiments for an epitaxial VS₂ monolayer. However, we apologize that to our experience a (close to) full monolayer of *quasi-freestanding* epitaxial VS₂ cannot be grown, irrespective of the substrate. For VS₂ on Au(111) the epitaxial orientation is excellent as found by Arnold *et al.*⁴, but the authors did not provide ARPES data. Presumably the strong hybridization of VS₂ with the substrate hinders the observation of clear bands characteristic for the monolayer. For the growth of quasi-freestanding stoichiometric VS₂ on Gr/Ir(111), the problem is not only the epitaxy. Good epitaxy can be achieved by higher annealing and/or growth temperature, but at the price of formation of sulfur-depleted and/or multilayer phases, similar to the observation of Arnold *et al.*⁴. Based on our extensive growth studies, it is thus not the growth temperature, but the inherent instability of stoichiometric VS₂, that limits the formation of coherent and epitaxial monolayers. The difficulties to grow stoichiometric monolayer islands are now clearly addressed in the first paragraph of the results section

During the revision of our manuscript, a manuscript of Kim *et al.*⁵ became available online describing epitaxial growth of VS₂ on bilayer Gr on SiC(0001) as suggested by the reviewer. However, although epitaxial, based on the V 3p core level signal obtained by x-ray photoelectron spectroscopy, stoichiometric VS₂ is a minority phase on the sample (cf. Fig. 2b, where the non-stoichiometric components V2 and V3 dominate). Together with the lack of evidence for monolayer rather than bilayer islands, this seems not to indicate that choosing bilayer Gr on SiC(0001) as substrate can be the solution.

Manuscript amended.

Reply to Reviewer 2

The authors investigate experimentally and theoretically the occurrence of charge density wave formation in single layer VS₂. The results are potentially interesting but there are a substantial number of points that should be clarified in order for this paper to be publication quality. The present version is not convincing and not suitable for publication in nature communications. The author should try to submit a revised version addressing the following critical points:

- 1. The use of the term “anharmonic” is misleading in all the paper. The authors use this term to state that, as the charge density wave phase involves large displacements with respect to the undistorted structure, there is a substantial admixture of transverse and longitudinal modes. However, they never calculate any anharmonic effect. Their free energy is harmonic (as far*

as I can understand from the paper, as there are no details) and their calculations in the CDW and non CDW phases are either total energy calculation or harmonic calculations. The free energy difference would be enormously affected by anharmonicity (calculation not included in this paper) and if anharmonicity were included the results would change substantially. What the authors mean is that as the CDW involves large atomic displacements, the CDW phase cannot be seen as a linear (small) perturbation of the undistorted phase and structural optimization must be performed. This is, by the way, not surprising as it is similar in other dichalcogenides, and explains why the Peierls picture breaks down. The large perturbation and the symmetry reduction of course allows for coupling of different modes. Thus, the authors should replace the term “anharmonic” (clearly misleading) with statements related to the non linear regime of the distortion.

The authors appreciate the suggestion to replace the term “anharmonic” by less ambiguous formulations, and we have updated our manuscript accordingly. We now refer to “non-linear mode–mode coupling” throughout the manuscript and included a new subsection, which deals with the non-linear regime of the distortion.

The term “anharmonic” is used in the literature essentially in two different ways. One is referring to calculations that map the anharmonic Born–Oppenheimer energy surface onto an effective harmonic potential, which in turn allows for the definition effective (temperature-dependent) phonons, which are renormalized by the “anharmonic effects” and accessible, e.g., via the self-consistent harmonic approximation.

We focus here, however, on the deviations of the underlying Born–Oppenheimer energy surface from the parabola of the harmonic approximation in general^{6–13}. In the present case, they lead to strong non-linear mode–mode coupling¹⁴ and deformation-induced changes on the electronic structure, which we identify in good agreement between theory and experiment.

The non-linear mode–mode coupling involving several different distinct wavevectors, as discussed here, is different from non-linearities or anharmonicities discussed in systems as like 2H-NbSe₂ or 2H-TaS₂: in the latter cases even very elaborate treatments of the CDW¹ usually consider a single effective phonon mode. That is not possible in the case of VS₂. To make the distinction between cases like 2H-TaS₂ and 1T-VS₂ clear, we added a comparison of corresponding Born–Oppenheimer energy surfaces to Fig. S5 of the SI and briefly discuss it in the main text.

Manuscript and SI amended.

- 2. The authors mention that “the total free energy is reduced by 24 meV per VS₂ unit”. It is not clear what the authors mean here. Is the total free energy the harmonic free energy including both the phonon (vibrational) contributions and the smearing (Fermi-Dirac) contribution? Are the author conscious that the statement that the free energy gain of 24 meV / VS₂ unit means an enormous energy gain in the supercell, order of magnitudes larger than what expected for a 300 K T_{CDW} (and much larger of what found in other single layer TMDs)? This should be rationalized a bit better, double checked and carefully compared with other calculations performed on 2D dichalcogenides.*

What is the origin of this gigantic energy gain at the PBE level ?

The authors should give a clear motivation of why this dichalcogenides is different from the others despite the practically identical electronic structure. Furthermore, the authors should also quote the total energy difference, so that we can distinguish if the difference arises from the vibrational part or not.

Finally, I am a bit confused by the harmonic free energy calculation for the undistorted phase. The harmonic free energy is well defined only for positive phonon frequencies as the resummation of the partition function requires the argument of the geometric series $\sum_n e^{-\beta\hbar\omega_n}$ to be smaller than one, otherwise the partition function diverges. How did the authors fix this problem in calculating the free energy difference between the two phases ? Could it be that this is at the origin of their large free energy difference ?

The authors apologize for not having been precise enough with their terminology in the previous version and thank the reviewer for pointing this out.

All energy differences mentioned in the manuscript are DFT “total energies” without vibrational contributions. Because of electronic smearing we originally referred to them as “free energies”. In order to avoid misunderstandings we use “total energy” in the revised manuscript. The abovementioned temperature effects are not topic of the paper at hand since we focus on properties of the Born–Oppenheimer energy surface alone.

The calculated total-energy gain per formula unit is indeed large compared to what is found for CDWs in *trigonal–prismatic* TMDCs, e.g., 3.7 meV per Nb atom for the 3×3 CDW in 2H-NbSe₂¹⁵. However, for the octahedral TMDCs these values are usually about one order of magnitude larger^{16,17}, e.g., 57 meV per Nb atom for the $\sqrt{13} \times \sqrt{13}$ CDW in 1T-NbSe₂¹⁸. Moreover, the calculated energy reduction for 1T-VS₂ is close to the 25 meV per V atom reported by Isaacs and Marianetti for the same system and similar DFT parameters¹⁹.

To put the total energy gains found here more clearly into context, we added a comparison to the aforementioned TMDC compounds to the section “Energetics of lattice instabilities” of the revised manuscript.

Manuscript amended.

- 3. The interesting XMCD data at the Vanadium L2,3 edges exclude the occurrence of a ferromagnetic ground state, as correctly noted by the authors. However, this does not exclude an antiferromagnetic ground state. The authors indeed found that the AF ground state is more stable in the CDW phases, however they did not characterize it (despite being the ground state!), mostly because they believe that calculations with other exchange and correlation kernels are needed. My feeling is that if the AF state is stable even at the PBE level, there are strong possibility that this is the real ground state. Thus, the authors should (i) calculate the energy gain of the AF-CDW state with respect to the (magnetic) ground state of the undistorted structure, (ii) calculate and quote the portion of the magnetic energy gain with respect to the structural energy gain, (iii) calculate the density of states of the CDW magnetic structure. How does it compare with the non magnetic one and with experiments ?*

The authors greatly appreciate the suggestions by the reviewer and performed the mentioned calculations for the $7 \times \sqrt{3}R30^\circ$ structure. In the following as well as in the updated manuscript, we will refer to the antiferromagnetic phase as the spin density wave (SDW).

Without CDW, the formation of the SDW reduces the total energy per formula unit by 7.1 meV. From there, the CDW displacement lowers the total energy by another 17.1 meV to 24.2 meV. This coupled CDW–SDW phase is by 1.5 meV more favorable than the CDW phase without SDW at 22.7 meV. These results are similar to previous calculations of ferromagnetism in the $9 \times \sqrt{3}R30^\circ$ structure of 1T-VS₂¹⁹ regarding both the large CDW energy gain and reduced SDW energy gain in the presence of the CDW.

The magnitude of the SDW energy gain is also reflected in the strength of the local magnetic moments. To illustrate this point, we updated Fig. 6b, c in the manuscript. We now show the SDW pattern both with and without CDW. With CDW (Fig. 6b), we find local moments up to only $\pm 0.18\mu_B$; without CDW in turn (Fig. 6c), the local moments reach $\pm 0.51\mu_B$. In the ferromagnetic case, Isaacs and Marianetti also find a reduction of local moments from about $0.5\mu_B$ to below $0.2\mu_B$ ¹⁹. There is thus a competition between the CDW lattice distortion and the formation of magnetic local moments.

We are also very thankful to the reviewer for the suggestion to analyze the influence of SDW formation on the electronic density of states (DOS). We follow this suggestion and added comparison of the DOS with and without SDW as Fig. S12 to the revised SI. Most importantly, the SDW leads to a reduction of the gap size. Taking the smearing of the experimental data into account, the DOS of the coupled CDW–SDW is in better agreement with the experiment regarding the shape of the gap than the non-spin polarized CDW DOS. This might be viewed as further indication that the SDW is realized in experiment on top of the CDW. We added a corresponding discussion to the revised manuscript.

Manuscript and SI amended.

4. *The graphical quality of the unfolded bands in blue in Fig. 4(a) is very poor (the same for the pictures in the supplemental materials) and does not seem to match the one of Fig. 4(c). For example, the Fermi surface in 4(c) exhibit a Fermi level crossing at approximately 1/2 of Γ –K. This crossing is invisible in Fig. 4(a). To me this is because the supercell breaks C3 and C6 symmetries and the different Γ –K directions of the undistorted structure Brillouin zone are not anymore equivalent when the bands are unfolded. In Fig. 4(a) the authors only pick one of these directions. Can the author improve the picture 4(a) so that to include also the other Γ –K direction with the elongated part of the cigar like Fermi surface crossing E_f ?*

The authors thank the reviewer for pointing out this possible source of ambiguity. The path through the Brillouin zone chosen in Fig. 5a in the manuscript is now indicated more prominently in Fig. 5b. In the manuscript, we chose to plot the portion of the band structure, where the opening of the gap is most obvious. The more complete representation requested by the reviewer can be found in Fig. S9 in the SI. We believe that the reduced figure is less overwhelming (especially now that we show three different CDW amplitudes) and thus more suitable for the main text. We also added

a comment regarding the broken C_3 symmetry and a reference to Fig. S9 to the caption of Fig. 5a.

Beyond that, we appreciate the reviewer’s criticism of the appearance of the unfolded bands, but unfortunately we do not see much room for improvement in this regard. Representing the spectral weight via the linewidth or mark size is common in the literature. Similar examples can be found, e.g., in Fig. 4d of Ref. 15 and Fig. 1a of Ref. 20, which show the unfolded electrons and phonons of 2H-NbSe₂.

Manuscript amended.

5. *Finally a crucial point. The authors underlined the opening of a band gap by the distortion in the unoccupied region. However, they give no explanation for it. Why the gap opens not at E_f but somewhere else ? This is a crucial point that the authors cannot avoid explaining.*

The authors are grateful that the reviewer raised this important issue. We extended both the manuscript and the SI to address the question of the position of the gap in due detail.

In the updated Fig. 5 and the Supplementary Videos, which show the electronic structure for an increasing amplitude of the lattice distortion, we can see that the gap indeed first opens at the Fermi level between M and K. With increasing displacements, also the (partial) gap grows, although this is not evident from the DOS, since it is “masked” by low-energy states near Γ , which always remain. It is not until the upper part of the gap surpasses the states near Γ that a complete gap becomes visible. Hence, the full gap we see above the Fermi level extends as a partial gap down below the Fermi level, i.e., it is not special in this regard. The energy gain is of course not associated to changes in the unoccupied states but only due to the simultaneous changes in the occupied states.

The question of why the complete gap is seen only above the Fermi level is thus largely equivalent to the question of why the states at Γ remain. We give an explanation based on electron count and symmetries as well as orbital structures and avoided crossings in Fig. S10 in the revised SI and refer to it in the revised manuscript in the section “Band structure and Fermi-surface topology”.

Manuscript and SI amended.

Reply to Reviewer 3

The physics driving the charge density wave (CDW) transition in solid-state crystals, and the role of Fermi surface nesting, have been longstanding subjects of debate within the solid-state community. The authors of this new manuscript make an important contribution to this discussion. Using a combination of scanning tunnelling microscopy/spectroscopy (STM/S), ab-initio calculation, and x-ray magnetic circular dichroism (XMCD), they argue compellingly that the CDW state in single layer VS₂/graphene/Ir(111) is driven by strong electron–phonon coupling, with little or no role for Fermi surface nesting. The paper presents many important new insights into the scope of the CDW phenomenon, the fundamental physics that underlie it, and its possible interplay with magnetism. The quality of the data is very good, and the calculations give good agreement with the experimental results. The findings are novel and will be of keen interest to the community, and are likely to stimulate future research in more than one direction:

1. “unconventional” CDW states associated with strong electron–phonon coupling and gaps opening away from the Fermi level—this latter being, to my knowledge, a topic which has not been widely recognised or understood before now;
2. interaction between CDWs and magnetism;
3. metal–metal Lifshitz transitions generated by electron–phonon-driven CDW transitions—an intriguing idea that only becomes possible once one moves beyond the Peierls’ view.

All three of these directions are “hot topics” within solid-state physics. Since there is no reason to think that the phenomena explored in the manuscript are limited to the particular material system treated in the study, or even to the broader category of two-dimensional materials, the work will be relevant to specialists spanning a wide spectrum of the solid-state research community.

I might also note that high-quality VS_2 is notoriously difficult to prepare, and the results presented in the manuscript point to very high sample quality. The authors have developed their growth method—using pyrite as a sulphur source—for some years, and have successfully realised a variety of single-layer transition metal dichalcogenides. However, the results presented here are, to my mind, particularly persuasive of the potential of the method for producing high-quality sulphide samples. Such results might stimulate larger-scale adoption of this method among other labs doing similar work.

Overall, I feel that the manuscript is appropriate to *Nature Communications*, on the basis of its

- novelty
- impact
- conceptual advances
- potential interest to readership
- important advances of significance to specialists within each field.

I do feel that there are some issues, however, that should be addressed in the manuscript before it is accepted for publication.

Sample stoichiometry: Previously published literature indicates that stoichiometric VS_2 can easily desulphidise into multiple S-deficient forms, both in the bulk and in the single-layer. Considering that STM does not give direct information about stoichiometry, some discussion of this issue is warranted. The authors convincingly defend their claim that the striped superstructure is the same stoichiometry as the non-striped structure. However, they do not (unless I am overlooking something) explicitly exclude the possibility that the stoichiometry of both the striped and non-striped phases could be something other than 1:2 V:S. (Note also the assertion at the top of p. 15 that STM was used to check the stoichiometry of their samples before XMCD measurements. This cannot be, strictly speaking, correct.) As a matter of fact, I do believe that the stoichiometry is 1:2, as the authors say, since the structure is hexagonal and highly ordered, with no visible defects in the atomically resolved data or di/dv mapping, and since agreement with theory is good. But I think this important issue should be discussed explicitly somewhere in the manuscript or the supplement.

The authors acknowledge the reviewer’s criticism on this issue. In fact, we have no inde-

pendent means to check the stoichiometry of the monolayer. We now discuss this issue properly in the first two paragraphs of the results section of the revised manuscript. We make clear that besides the lattice symmetry, the proper lattice constant, the reversible switching between the wave pattern and the plain 1×1 lattice also the analogy to the observations of Arnold *et al.*⁴ with respect to sulfur depletion and stoichiometry underpin our conclusion that the observed phase is stoichiometric. In fact, similar to Arnold *et al.*⁴, also we observe S-depleted phases after annealing to higher temperatures, as is now explicitly pointed out. The assignment “stoichiometric” is part of the concluding last sentence of the first two paragraphs, after all arguments were introduced. Thereby the reader is not forced to accept “stoichiometric,” but can judge on how reasonable our conclusion is, based on all arguments explicated.

Manuscript amended.

Discussion of gap: *Naïvely speaking, it seems surprising that the CDW gap in Fig. 3 is already closing at 78.5 K, considering that the CDW state persists to >300 K. Can the authors explain this?*

In the revised manuscript, at the description of the spectra we point out the possible reasons for its absence at 78.5 K (loss of energy resolution, reduction of distortions with temperature). We take up this topic again, when discussing the calculated band structure, where it becomes apparent that the size and existence of the gap depend sensitively on the overlap of two gaps and their distortions. Spin-polarized DFT calculations have additionally revealed that a spin density wave, as expected for the anti-ferromagnetic ground state, would have a considerably smaller gap. In that case, with the gap already quite small at 7 K, the difference between 7 K and 78.5 K might lie for the most part in loss of energy resolution.

Manuscript amended.

Discussion of V atom displacement: *The displacement of the V atoms from equilibrium is estimated to be 8%, on the basis of theory. The authors note (p. 10) that this might be an overestimate. In any case, 8% strikes me as huge. Can the authors comment on this? How does this compare to CDW distortions in other material systems? How sensitively dependent are calculated densities of states and electronic dispersions upon this value? This seems like an important issue that deserves some discussion, even if only in the supplement.*

The authors thank the reviewer for raising this important issue. A displacement of the order of 8% is large compared to equivalent values observed in the trigonal–prismatic TMDCs, but it is not unexpected for octahedral TMDCs^{16,17}. For comparison, the calculated maximum displacement in the CDW of 2H-NbSe₂ (3×3) and 1T-NbSe₂ ($\sqrt{13} \times \sqrt{13}$) is 2.3%¹⁵ and 8.8%¹⁸ of the lattice constant, respectively. We give a comparison of distortions in 1T-VS₂ to these other TMDC compounds in the revised manuscript.

The dependence of the Fermi surface, band structure, and DOS on the displacement amplitude can be seen in the Supplementary Videos. To address this important point also directly in the main text, we updated Fig. 5 in the manuscript and show the Fermi surface for different displacement

amplitudes. From the Supplementary Video 1, we learn that the complete gap above the Fermi level starts to emerge at 70 % of the final displacements, i.e., at a maximum displacement of 5.7 % of the lattice constant. DFT+ U calculations¹⁹ and also our spin-polarized DFT calculations added in revised Fig. S12 in the SI show that the effects of distortions and local magnetic moments in 1T-VS₂ counteract regarding the gap opening in the DOS. Thus, the experimental observation of a full gap strongly suggest that the real displacements do not fall much below the calculated ones. A corresponding comment has been added to the revised main text.

Manuscript amended.

Caption to Fig. 2: *I am not sure to what “colored stripes in the background” refers. Are these the darker and lighter yellow regions?*

With “colored striped in the background” we indeed referred to these darker and lighter regions. We have amended the manuscript and now use a more reasonable representation of the charge-density wavelength through a color gradient.

Manuscript amended.

Figs. 1, 3: *I do not see the “indigo box” that is referenced in the caption to Fig. 3, nor do I see indigo circles in the Fourier transforms. I do see grey boxes and grey circles—is it these to which the caption refers? In general, the grey is somewhat difficult to see, both in Figs. 1 & 3. The grey arrows in the insets (Fourier transforms) in panels (b) and (c) are particularly difficult to see.*

The authors apologize for these errors. We replaced the grey boxes in Fig. 1 and 3 by blue boxes that are better visible. We also removed the box in Fig. 1c. The text and the caption are updated accordingly.

Manuscript amended.

P. 6: *The text reads: “For the DFT calculations below, the experimental wave pattern must be approximated by a commensurate supercell. Two close approximations are overlaid on the atomic resolution images in Fig. 1(b, c). The gray boxes in Fig. 1(b) and (c) indicate $9 \times \sqrt{3}R30^\circ$ and $7 \times \sqrt{3}R30^\circ$ structure with periodicities $(2.25 \pm 0.02)a_{\text{VS}_2}$ and $(2.33 \pm 0.02)a_{\text{VS}_2}$, respectively.”*

My understanding is that the observed superstructure is neither $9 \times \sqrt{3}R30^\circ$ nor $7 \times \sqrt{3}R30^\circ$, but is rather an incommensurate structure falling between these two, and that the commensurate structures are introduced for the sake of computational tractability. Is that correct? However, in Fig. 1, the $9 \times \sqrt{3}R30^\circ$ structure is superimposed on the data acquired at 7 K in panel (b), whereas the $7 \times \sqrt{3}R30^\circ$ structure is superimposed on the 300 K data in panel (c)—which would seem to suggest that the two commensurate structures are temperature-dependent phases. Perhaps the authors could change the presentation of the structural models, to avoid confusion.

The authors are grateful for the criticism on the structure models. Indeed, the introduction of

the unit cells on topographs taken at different temperatures may be misleading. For simplicity, we removed the $7 \times \sqrt{3}R30^\circ$ overlay from Fig. 1c. The main text explicitly mentions now the absence of a temperature dependence of the CDW vector. It also emphasizes that the two unit cells are only introduced for computational purposes. Deviations in the computational results allow to judge how far these results might deviate from the incommensurate situation with a CDW vector tightly in between.

Manuscript amended.

References

1. Flicker, F. & van Wezel, J. Charge order from orbital-dependent coupling evidenced by NbSe₂. *Nat. Commun.* **6**, 7034 (2015). URL <https://doi.org/10.1038/ncomms8034>.
2. Chen, P. *et al.* Unique gap structure and symmetry of the charge density wave in single-layer VSe₂. *Phys. Rev. Lett.* **121**, 196402 (2018). URL <https://doi.org/10.1103/PhysRevLett.121.196402>.
3. Feng, J. *et al.* Electronic structure and enhanced charge-density wave order of monolayer VSe₂. *Nano Lett.* **18**, 4493 (2018). URL <https://doi.org/10.1021/acs.nanolett.8b01649>.
4. Arnold, F. *et al.* Novel single-layer vanadium sulphide phases. *2D Mater.* **5**, 045009 (2018). URL <https://doi.org/10.1088/2053-1583/aad0c8>.
5. Kim, H. J. *et al.* Electronic structure and charge-density wave transition in monolayer VS₂. *Curr. Appl. Phys.* (2021). URL <https://doi.org/10.1016/j.cap.2021.03.020>.
6. Cowley, R. A. Structural phase transitions I. Landau theory. *Advances in Physics* **29**, 1 (1980). URL <https://doi.org/10.1080/00018738000101346>.
7. Varma, C. M. & Simons, A. L. Strong-coupling theory of charge-density-wave transitions. *Phys. Rev. Lett.* **51**, 138 (1983). URL <https://doi.org/10.1103/PhysRevLett.51.138>.
8. Withers, R. L. & Wilson, J. A. An examination of the formation and characteristics of charge-density waves in inorganic materials with special reference to the two- and one-dimensional transition-metal chalcogenides. *J. Phys. C: Solid State Phys.* **19**, 4809 (1986). URL <https://doi.org/10.1088/0022-3719/19/25/005>.
9. Mankowsky, R. *et al.* Nonlinear lattice dynamics as a basis for enhanced superconductivity in YBa₂Cu₃O_{6.5}. *Nature* **516**, 71 (2014). URL <https://doi.org/10.1038/nature13875>.
10. Leonov, I. *et al.* Electronic correlations determine the phase stability of iron up to the melting temperature. *Sci. Rep.* **4**, 5585 (2014). URL <https://doi.org/10.1038/srep05585>.

11. Skelton, J. M. *et al.* Anharmonicity in the high-temperature $Cmcm$ phase of SnSe: Soft modes and three-phonon interactions. *Phys. Rev. Lett.* **117**, 075502 (2016). URL <https://doi.org/10.1103/PhysRevLett.117.075502>.
12. Wall, S. *et al.* Ultrafast disordering of vanadium dimers in photoexcited VO₂. *Science* **362**, 572 (2018). URL <https://doi.org/10.1126/science.aau3873>.
13. Jong, U.-G. *et al.* Anharmonic phonons and phase transitions in the vacancy-ordered double perovskite Cs₂SnI₆ from first-principles predictions. *Phys. Rev. B* **99**, 184105 (2019). URL <https://doi.org/10.1103/PhysRevB.99.184105>.
14. Truitt, P. A., Hertzberg, J. B., Altunkaya, E. & Schwab, K. C. Linear and nonlinear coupling between transverse modes of a nanomechanical resonator. *J. Appl. Phys.* **114**, 114307 (2013). URL <https://doi.org/10.1063/1.4821273>.
15. Lian, C.-S., Si, C. & Duan, W. Unveiling charge-density wave, superconductivity, and their competitive nature in two-dimensional NbSe₂. *Nano Lett.* **18**, 2924 (2018). URL <https://doi.org/10.1021/acs.nanolett.8b00237>.
16. Rossnagel, K. On the origin of charge-density waves in select layered transition-metal dichalcogenides. *J. Phys. Condens. Matter* **23**, 213001 (2011). URL <https://doi.org/10.1088/0953-8984/23/21/213001>.
17. Miller, D. C., Mahanti, S. D. & Duxbury, P. M. Charge density wave states in tantalum dichalcogenides. *Phys. Rev. B* **97**, 045133 (2018). URL <https://doi.org/10.1103/PhysRevB.97.045133>.
18. Calandra, M. Phonon-assisted magnetic Mott-insulating state in the charge density wave phase of single-layer 1T-NbSe₂. *Phys. Rev. Lett.* **121**, 026401 (2018). URL <https://doi.org/10.1103/PhysRevLett.121.026401>.
19. Isaacs, E. B. & Marianetti, C. A. Electronic correlations in monolayer VS₂. *Phys. Rev. B* **94**, 035120 (2016). URL <https://doi.org/10.1103/PhysRevB.94.035120>.
20. Zheng, F. & Feng, J. Electron-phonon coupling and the coexistence of superconductivity and charge-density wave in monolayer NbSe₂. *Phys. Rev. B* **99**, 161119 (2019). URL <https://doi.org/10.1103/PhysRevB.99.161119>.

REVIEWERS' COMMENTS

Reviewer #1 (Remarks to the Author):

I thank the authors for their comprehensive reply to my queries. I am satisfied with the response and have no further objections.

Reviewer #3 (Remarks to the Author):

The authors have improved the manuscript with substantive changes and additions. They have addressed my concerns. I find the manuscript now to be appropriate for publication. I have one remaining question, in regard to the newly added material, and two very minor additional points:

Fig. S11: the sample shows 40% coverage, which consists of a combination of trilayer and monolayer regions. This is not surprising in this sample system. Can they estimate the ratio of trilayer (or thicker) coverage to monolayer coverage? What impact do they believe the presence of trilayer regions has on the XMCD results?

Fig. 1a: the temperature is (unless I am mistaken) not explicitly stated for this panel.

Fig. 2a: the interpretation of the inset Brillouin zone schematic is unclear to me. What do the dark- and light-grey panels signify?

Response to referees

1. We thank the reviewer for addressing this interesting question. We estimate that the sample has about 75% monolayer and 25% multilayer (a combination of bilayer and trilayer). The robust absence of any magnetism in the XMCD measurement leads us to conclude that the multilayer either had a negligible contribution to the magnetic signal, or that it is also not ferro- or paramagnetic. The magnetic, electronic and structural properties of multilayer VS₂ remain an open question at this point.

We have amended to manuscript so that below figure S11 it reads: 'Distinct height levels indicate up to three layers, with multilayer VS₂ making up 25% of the total amount of VS₂ present on the surface.'

2. Caption of figure 1a now mentions the temperature at which the STM topograph was taken (7K). Manuscript amended.

3. We thank the reviewer for pointing out that the interpretation of the inset is unclear. We show the phonon dispersion along a path that leaves the first Brillouin zone to emphasize the swapped LA/TA mode character at the M and M' points. The dark- and light-grey panels represent the twelve irreducible wedges of the Brillouin zone and are intended to provide guidance in this extended zone scheme. For clarity, we now additionally indicate the Brillouin-zone boundaries using solid black lines.